# The deep-subsurface sulfate reducer *Desulfotomaculum kuznetsovii* employs two methanol-degrading pathways

Diana Z. Sousa[1], Michael Visser [1], Antonie H. van Gelder[1], Sjef Boeren [2], Mervin M. Pieterse[3,4], Martijn W.H. Pinkse[3,4], Peter D.E.M. Verhaert [3,4,5,6], Carsten Vogt[7], Steffi Franke[7], Steffen Kümmel[7] & Alfons J.M. Stams[1,8]

Methanol is generally metabolized through a pathway initiated by a cobalamin-containing methanol methyltransferase by anaerobic methylotrophs (such as methanogens and acetogens), or through oxidation to formaldehyde using a methanol dehydrogenase by aerobes. Methanol is an important substrate in deep-subsurface environments, where thermophilic sulfate-reducing bacteria of the genus *Desulfotomaculum* have key roles. Here, we study the methanol metabolism of *Desulfotomaculum kuznetsovii* strain 17[T], isolated from a 3000-m deep geothermal water reservoir. We use proteomics to analyze cells grown with methanol and sulfate in the presence and absence of cobalt and vitamin B12. The results indicate the presence of two methanol-degrading pathways in *D. kuznetsovii*, a cobalt-dependent methanol methyltransferase and a cobalt-independent methanol dehydrogenase, which is further confirmed by stable isotope fractionation. This is the first report of a microorganism utilizing two distinct methanol conversion pathways. We hypothesize that this gives *D. kuznetsovii* a competitive advantage in its natural environment.

[1] Laboratory of Microbiology, Wageningen University & Research, Stippeneng 4, 6708 WE Wageningen, The Netherlands. [2] Laboratory of Biochemistry, Wageningen University & Research, Stippeneng 4, 6708 WE Wageningen, The Netherlands. [3] Department of Biotechnology, Delft University of Technology, Julianalaan 67, 2628 BC Delft, The Netherlands. [4] Netherlands Proteomics Centre, Julianalaan 67, 2628 BC Delft, The Netherlands. [5] M4i, Maastricht Multimodal Molecular Imaging Institute, Faculty of Health, Medicine & Life Sciences, University of Maastricht, 6229 ER Maastricht, The Netherlands. [6] ProteoFormiX, Janssen Pharmaceutica Campus, B2340 Beerse, Belgium. [7] Department of Isotope Biogeochemistry, UFZ-Helmholtz Centre for Environmental Research, Permoserstraße 15, 04318 Leipzig, Germany. [8] Centre of Biological Engineering University of Minho, Campus de Gualtar, 4710-057 Braga, Portugal. Diana Z. Sousa and Michael Visser contributed equally to this work. Correspondence and requests for materials should be addressed to A.J.M.S. (email: fons.stams@wur.nl)

High temperatures and oligotrophic conditions often prevail in deep-subsurface environments, which can be useful for underground gas storage and geothermal energy production[1]. However, the resident microbial communities influence possible applications, and these in turn affect the ecology of the deep-subsurface microbiota. Therefore, understanding the microbial composition of deep-subsurface environments and the metabolism of their community members is important. Studies so far showed a dominance of Gram-positive, spore forming, thermophilic bacteria in high-temperature subsurface environments, especially *Desulfotomaculum* species[2–5]. Many *Desulfotomaculum* species are thermophilic and can grow in vitamin-deprived environments[6,7]. They possess a rather versatile metabolism and their spores are extremely heat resistant[8,9], which make them perfectly adapted to subsurface conditions.

Methanol is an important substrate for microbial life in deep-subsurface environments[10,11]. Methanol is a common compound in nature and it is naturally produced by the degradation of pectin and lignin, which are constituents of plant cell walls[12]. However, in the deep-subsurface methanol may be geochemically produced from $CO_2$ and $H_2$, a gas mixture commonly present in these environments due to the geological production of hydrogen. Abiotic synthesis of methanol in conditions characteristic for deep-subsurface environments was described[13].

Several phylogenetic groups of microorganisms are able to grow with methanol as a sole carbon and energy source. Aerobic and facultative anaerobic methylotrophs generally convert methanol to formaldehyde by a methanol dehydrogenase (MDH). Multiple MDHs, such as MDHs that use pyrroloquinoline quinone (PQQ) or NAD(P) as a cofactor, have been characterized[14,15]. Recently, two types of PQQ-dependent MDHs were described to be present in *Methylobacterium extorquens* AM1. A PQQ MDH using calcium in its active site and another using lanthanides[16]. In anoxic deep-subsurface environments methylotrophs such as methanogenic archaea, acetogenic bacteria, and sulfate-reducing bacteria compete for methanol. Methanogens and acetogens employ a methanol methyltransferase (MT) system[17–23]. This system involves two enzymes, $MT_1$ and $MT_2$. $MT_1$ consists of two subunits, the first (MtaB) is involved in breaking the C–O bond of methanol and transferring the methyl residue to the second subunit (MtaC). $MT_2$ (MtaA) transfers the methyl group from MtaC to coenzyme M in methanogens[17–20], or tetrahydrofolate in acetogens[21–23].

The methanol metabolism of sulfate-reducing bacteria (SRB) has not been extensively studied. It is not clear whether SRB use a MT system or a MDH. Several SRB utilize methanol for growth, such as *Desulfosporosinus orientis*[24], *Desulfobacterium catecholicum*[25], *Desulfobacterium anilini*[26], *Desulfovibrio carbinolicus*[27], *Desulfovibrio alcoholivorans*[28], and nine *Desulfotomaculum* strains[10,29–33] including *D. kuznetsovii*. The latter species is a methylotrophic thermophilic sulfate-reducing bacterium that was isolated from a geothermal water reservoir at a depth of about 3000 m[10]. We studied the metabolism of this sulfate reducer to get insight into its growth strategy in oligotrophic deep-subsurface environments. Growth of *D. kuznetsovii* with methanol and sulfate was studied and resulted in a partially purified alcohol dehydrogenase (ADH) with a molecular mass of 42 kDa that also showed activity with methanol, but activity with ethanol was ten times higher[34]. Analysis of the genome of *D. kuznetsovii* revealed the putative presence of methanol methyltransferase genes as well[7]. Therefore, the methanol metabolism in *D. kuznetsovii* remained unsolved and we hypothesized that the bacterium possesses two distinct methanol-degradation pathways, which has never been described in other microorganisms.

Here we show evidence for the presence of two methanol-degradation pathways in *D. kuznetsovii* by analyzing the proteome of cells grown with methanol and sulfate in the presence and absence of cobalt and vitamin B12. Importantly, stable isotope fractionation analysis of cells grown in media with cobalt and vitamin B12 indicates that during growth the alcohol dehydrogenase is used first and the MT is operating later at lower methanol concentrations.

## Results

**Effect of cobalt and vitamin B12 on growth with methanol.** The presence of genes coding for a methanol MT system in the genome of *D. kuznetsovii* suggested the involvement of a vitamin B12-dependent MT system in methanol conversion[7], while previous analysis indicated the involvement of an alcohol dehydrogenase[34]. To clarify the role of these enzyme systems we assessed the effect of cobalt on growth with methanol.

When cobalt and vitamin B12 were omitted from the medium *D. kuznetsovii* was still able to degrade methanol, but the residual methanol concentration at the end of the assays was significantly higher ($p = 0.00027$) than in assays with cobalt and vitamin B12 (Supplementary Fig. 1). This indicates the presence of a second, cobalamin-independent, methanol-degradation pathway, and suggests the importance of the methanol MT system for the conversion of low concentrations of methanol.

**Comparative proteomics shows two methanol-degrading pathways.** *D. kuznetsovii* cells were adapted to four different growth conditions: methanol and sulfate in presence and absence of cobalt and vitamin B12, lactate and sulfate, and ethanol and sulfate. The lactate growth condition was used as a reference, whereas the ethanol growth condition was used because previous research indicated the involvement of an alcohol dehydrogenase for growth with methanol and ethanol[34]. Protein abundance data under the different conditions are shown in Supplementary Data 1, and the abundance of physiologically important proteins involved in methanol metabolism is visualized in Fig. 1. Assays with methanol were performed using initial substrate concentrations of 20 mM and 5 mM, but main results and trends were similar for both conditions (for this reason results from assays with 5 mM methanol are omitted in the manuscript and provided only in Supplementary Data 1).

Growth of *D. kuznetsovii* with methanol in the presence of cobalt and vitamin B12 resulted in increased abundance of proteins encoded by genes of an operon (Desku_0050-60), which were annotated as proteins involved in vitamin B12 biosynthesis and a predicted methanol MT system (Fig. 1; Supplementary Data 1). Two MtaA MTs, a MtaB and MtaC are highly abundant under these conditions. The increased abundance of the corrinoid binding MtaC indicates the necessity of vitamin B12 in the cell. No vitamin B12 transport encoding genes were found in the genome of *D. kuznetsovii* and all genes essential for vitamin B12 synthesis were present in the genome[7]. Only vitamin B12 synthesis proteins encoded by genes of the operon structure Desku_0050–0060 were more abundant during growth with methanol and cobalt, which coincides with the higher expression of the MT system in these conditions.

In other studies, cobalt limitation led to decreased conversion rates of methanogens and acetogens when grown with methanol[35–38]. This was explained by the essential role of cobalt in corrinoid biosynthesis[38] and the synthesis of corrinoid-dependent proteins by the methanol utilizers[20,35–37,39]. The MtaC subunit of the methanol MT system was described to bind the corrinoid[21,40,41]. When cobalt and vitamin B12 were omitted from the medium the abundance of the MT system and the vitamin B12 synthesis pathway were very low (Fig. 1). Growth on methanol (with and without cobalt and vitamin B12) and on

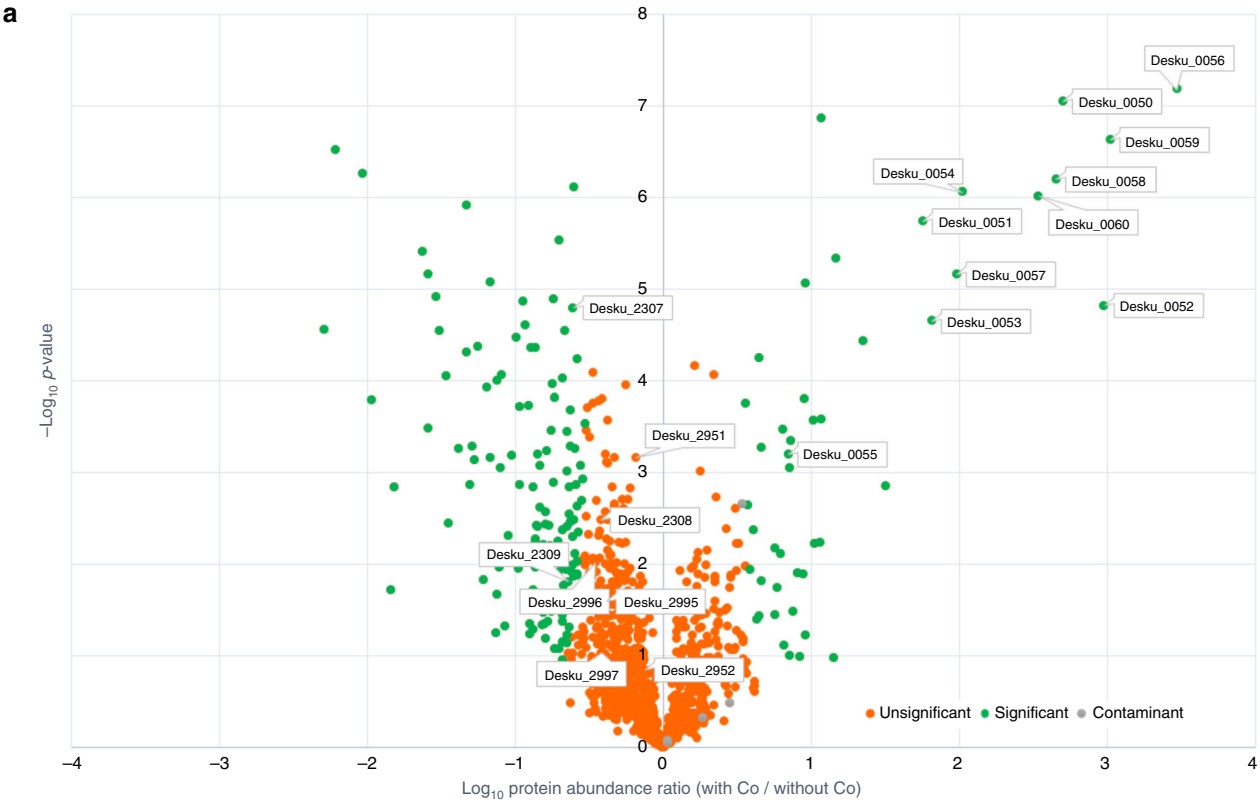

| Gene name | Uniprot | Protein name | EtOH control | | | Lactate control | | | MeOH with Co | | | | MeOH without Co | | | |
|---|---|---|---|---|---|---|---|---|---|---|---|---|---|---|---|---|
| | | | 1 | 2 | 3 | 1 | 2 | 3 | 1 | 2 | 3 | 4 | 1 | 2 | 3 | 4 |
| Desku_0050 | F6CLM4 | Methyltransferase MtaA/CmuA family | 6.2 | 5.9 | 5.4 | 5.1 | 5.7 | 6.4 | 9.6 | 9.5 | 9.4 | 9.2 | 6.8 | 6.8 | 6.7 | 6.6 |
| Desku_0051 | F6CLM5 | Methanol:cobalamin methyltransferase, subunit B | 8.3 | 8.0 | 8.0 | 7.8 | 7.7 | 8.0 | 9.8 | 9.8 | 9.8 | 10.0 | 8.3 | 8.1 | 8.1 | 7.9 |
| Desku_0052 | F6CLM6 | Methyltransferase cognate corrinoid protein | 5.9 | 6.4 | 5.8 | 6.1 | 6.0 | 7.1 | 9.2 | 9.2 | 9.2 | 9.3 | 6.9 | 6.1 | 6.3 | 5.8 |
| Desku_0053 | F6CLM7 | Cobalamin synthesis protein P47K | 5.7 | 6.1 | 5.8 | 5.7 | 6.0 | 5.0 | 7.6 | 7.5 | 7.5 | 7.7 | 6.1 | 5.8 | 5.5 | 5.6 |
| Desku_0054 | F6CLM8 | 4Fe-4S ferredoxin iron-sulfur binding domain-containing protein | 5.9 | 6.3 | 5.7 | 5.9 | 6.1 | 5.8 | 7.8 | 7.7 | 7.9 | 8.1 | 5.8 | 5.9 | 5.8 | 5.9 |
| Desku_0055 | F6CLM9 | Uroporphyrinogen decarboxylase (URO-D) | 6.0 | 6.2 | 5.8 | 5.9 | 6.0 | 6.0 | 6.9 | 6.9 | 6.7 | 6.8 | 6.1 | 5.6 | 6.0 | 6.1 |
| Desku_0056 | F6CLN0 | Methionine synthase B12-binding module cap domain protein | 5.3 | 5.7 | 6.0 | 5.7 | 5.8 | 5.9 | 9.1 | 9.1 | 9.1 | 8.7 | 5.3 | 5.6 | 5.6 | 5.4 |
| Desku_0057 | F6CLN1 | Ferredoxin | 8.0 | 8.0 | 6.9 | 7.0 | 6.9 | 7.2 | 9.4 | 9.4 | 9.3 | 9.4 | 7.8 | 7.1 | 7.3 | 7.3 |
| Desku_0058 | F6CLN2 | Tetrahydromethanopterin S-methyltransferase | 6.9 | 6.4 | 6.4 | 6.2 | 6.0 | 6.3 | 9.2 | 9.0 | 9.1 | 9.2 | 6.7 | 6.3 | 6.3 | 6.5 |
| Desku_0059 | F6CLN3 | Pyridoxamine 5-phosphate oxidase-related FMN-binding protein | 5.9 | 5.8 | 6.2 | 5.5 | 6.7 | 6.2 | 9.4 | 9.4 | 9.2 | 9.4 | 6.3 | 6.4 | 6.0 | 6.5 |
| Desku_0060 | F6CLN4 | Methyltransferase MtaA/CmuA family | 7.3 | 6.9 | 6.8 | 6.4 | 6.4 | 6.8 | 9.7 | 9.6 | 9.6 | 9.8 | 7.1 | 6.9 | 7.4 | 7.2 |
| Desku_2951 | F6CJW1 | Aldehyde ferredoxin oxidoreductase | 10.3 | 10.3 | 10.4 | 9.1 | 9.2 | 9.3 | 10.3 | 10.3 | 10.3 | 10.3 | 10.4 | 10.5 | 10.5 | 10.5 |
| Desku_2952 | F6CJW2 | 1,3-propanediol dehydrogenase | 9.6 | 9.7 | 9.6 | 8.0 | 7.9 | 8.0 | 10.5 | 10.4 | 10.5 | 10.2 | 10.5 | 10.5 | 10.5 | 10.5 |
| Desku_2307 | F6CNN4 | Hydrogenase, Fe-only | 8.5 | 8.3 | 8.4 | 7.1 | 7.3 | 7.4 | 9.1 | 9.0 | 9.2 | 9.0 | 9.7 | 9.8 | 9.7 | 9.6 |
| Desku_2308 | F6CNN5 | NADH dehydrogenase (quinone) | 8.2 | 8.1 | 8.3 | 7.6 | 7.5 | 7.6 | 9.5 | 9.1 | 9.3 | 9.2 | 9.6 | 9.7 | 9.7 | 9.7 |
| Desku_2309 | F6CR85 | Ferredoxin | 7.4 | 7.3 | 7.3 | 5.1 | 5.8 | 6.2 | 8.4 | 8.4 | 8.5 | 7.9 | 9.0 | 9.0 | 9.1 | 8.6 |
| Desku_2995 | F6CKC7 | Hydrogenase, Fe-only | 9.9 | 9.9 | 9.9 | 9.4 | 9.5 | 9.5 | 8.4 | 8.7 | 8.5 | 8.7 | 8.9 | 8.7 | 9.3 | 9.0 |
| Desku_2996 | F6CKC8 | NADH dehydrogenase (quinone) | 9.8 | 9.6 | 9.7 | 9.2 | 9.2 | 9.4 | 8.3 | 8.4 | 8.2 | 8.6 | 8.8 | 8.7 | 9.1 | 8.7 |
| Desku_2997 | F6CKC9 | NADH dehydrogenase (quinone) | 8.9 | 8.7 | 9.0 | 7.9 | 8.2 | 8.1 | 7.1 | 7.8 | 7.7 | 7.6 | 8.0 | 7.7 | 8.4 | 7.8 |

Low　　　　　　　　High

**Fig. 1** Comparative proteomics results. **a** Volcano plot with comparison of cells grown on 20 mM methanol with and without supplementation of cobalt and vitamin B12. Data are from four independent replicates (Supplementary Data 1). **b** Identification of the predicted function of proteins depicted in the volcano plot and corresponding label-free quantification (LFQ) values for proteins quantified in cells grown with different electron donors (20 mM ethanol, 20 mM ethanol, 20 mM methanol with and without supplementation of cobalt, and vitamin B12)

ethanol resulted in high abundance of an alcohol dehydrogenase (Desku_2952) and an aldehyde ferredoxin oxidoreductase (Desku_2951) (Fig. 1), indicating the involvement of those proteins in both the methanol and ethanol metabolism of *D. kuznetsovii*.

Goorissen partially purified an ADH with a molecular mass of 42 kDa that showed activity with ethanol and methanol[34]. In that study, the ADH was present during growth with ethanol and sulfate, but was more abundant during growth with methanol and sulfate. However, the ADH activity with ethanol was ten times higher than with methanol. Activity could be measured with nicotinamide adenine dinucleotide (NAD), 2,6 dichlorophenolindophenol (DCPIP), and 3-(4,5-dimethylthiazol-2-yl)-2,4 diphenyltetrazolium bromide (MTT), but not with nicotinamide

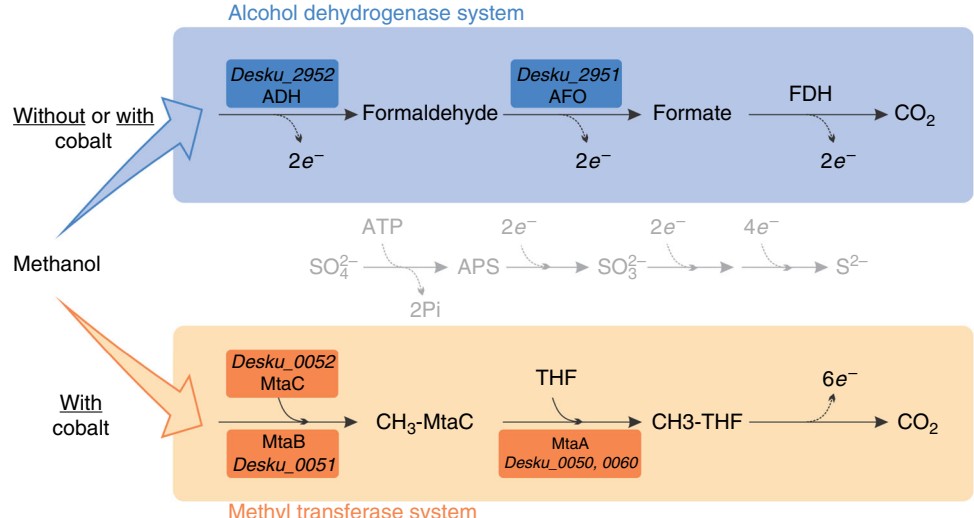

**Fig. 2** Hypothesized methanol metabolism pathways in *D. kuznetsovii*. Methanol is oxidized to $CO_2$ by an alcohol dehydrogenase (ADH), aldehyde ferredoxin oxidoreductase (AFO), and a formate dehydrogenase (FDH). When cobalt is present in the environment a second concurrent methanol-oxidizing pathway is induced and part of the methanol is methylated to methyl-tetrahydrofolate (CH₃-THF). Subsequently, CH₃-THF is oxidized to $CO_2$ generating the same amount of electrons. Locus tag numbers are indicated for boxed enzymes

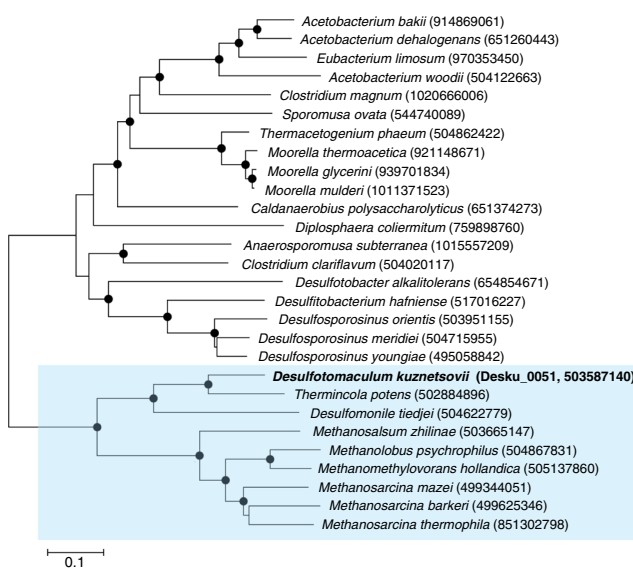

**Fig. 3** Neighbor-joining tree based on MtaB amino acid sequences. The sequences were obtained from a BLASTp analysis, using MtaB of *D. kuznetsovii* as the query sequence. MtaB of *D. kuznetsovii* is printed in bold. Closed circles represent bootstrap values of 75% or higher. Scale bar represents 10% sequence difference

adenine dinucleotide phosphate (NADP). The highest activity was measured with ethanol and NAD. Moreover, activity of the reverse reaction was measured when using both acetylaldehyde and formaldehyde[34].

Our results indicate that the partially purified ADH described by Goorissen is the Desku_2952 ADH. In agreement with that study the abundance of the Desku_2952 ADH is higher when cells were grown with methanol compared to ethanol-grown cells (Fig. 1b) and the predicted size of the Desku_2952 ADH is 41 kDa. Two other alcohol dehydrogenases (Desku_0619, 3082) and four other aldehyde dehydrogenases (Desku_0621, 2946, 2983, 3081) were identified in the genome and some in proteome data

(Supplementary Data 1), but these did not exhibit enhanced abundance in any of the growth conditions that we tested or any abundance at all. Therefore, they do not seem to be specifically involved in the ethanol and/or methanol degradation. These results suggest that two methanol-utilizing pathways are present in *D. kuznetsovii* as visualized in Fig. 2.

The MtaB (Desku_0051) and the ADH (Desku_2952) amino acid sequences and closely related protein sequences of other microorganisms were used to generate phylogenetic trees (Figs. 3 and 4). Figure 3 shows the distribution of MtaB proteins of sulfate reducers, acetogens, and methanogens. Interestingly, the phylogenetic tree displays two major clades where *D. kuznetsovii* resides in the same clade as methanogens, while other Gram-positive SRB, like *Desulfosporosinus* species, cluster together with acetogens in the other clade. This leads to the suggestion that the MT system of *D. kuznetsovii* is evolutionarily closer to the MT system of methanogens than to that of acetogens, which is a remarkably unexpected finding. This could be due to a horizontal gene transfer event.

*D. kuznetsovii* has six ADH encoding genes in its genome, which cluster separately in an amino acid sequence neighbor-joining tree (Fig. 4). This suggests that their sequences differ from each other, which could coincide with different substrate specificity. Interestingly, the methanol-oxidizing ADH clusters together with ADH sequences of species that are able to use ethanol, but are unable or not known to utilize methanol.

**Stable isotope fractionation analysis.** The proteomics data showed that enzymes of the two methanol-degrading pathways are produced when *D. kuznetsovii* is grown with methanol and sulfate in the presence of cobalt. To assess the contribution of each pathway under these conditions we performed a compound specific stable carbon isotope analysis. The methyltransferase reaction has been shown to result in a large stable carbon isotope fractionation upon methanol conversion to methane by methanogens[42,43]. No data are available for carbon isotope fractionation of methanol oxidation catalyzed by an ADH. The rate-limiting step upon methanol oxidation of the PQQ-depending ADH is assumed to be the breakage of the methyl C–H bond, leading to a large deuterium isotope effect[14], but this step is not linked to

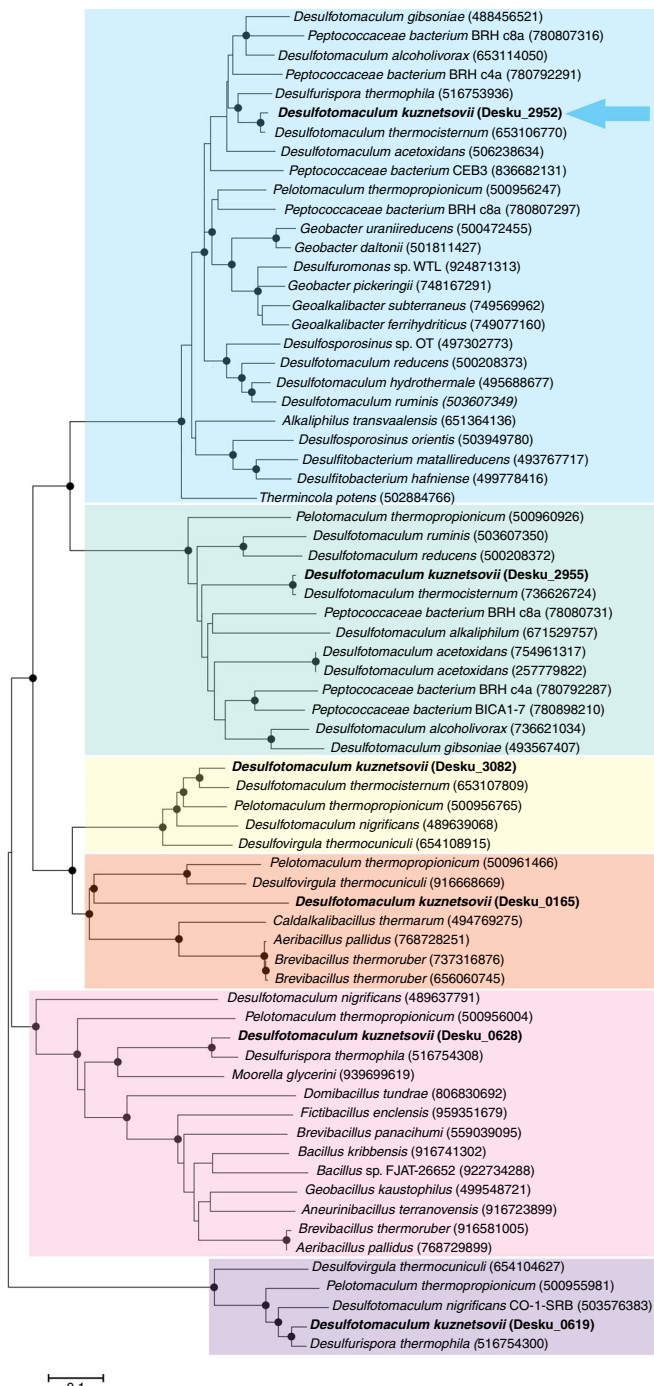

**Fig. 4** Neighbor-joining tree based on ADH amino acid sequences. The sequences were obtained from a BLASTp analysis, using ADHs of *D. kuznetsovii* as query sequences. ADHs of *D. kuznetsovii* are printed in bold and an arrow points at the methanol-oxidizing ADH. Closed circles represent bootstrap values of 75% or higher. Scale bar represents 10% sequence difference

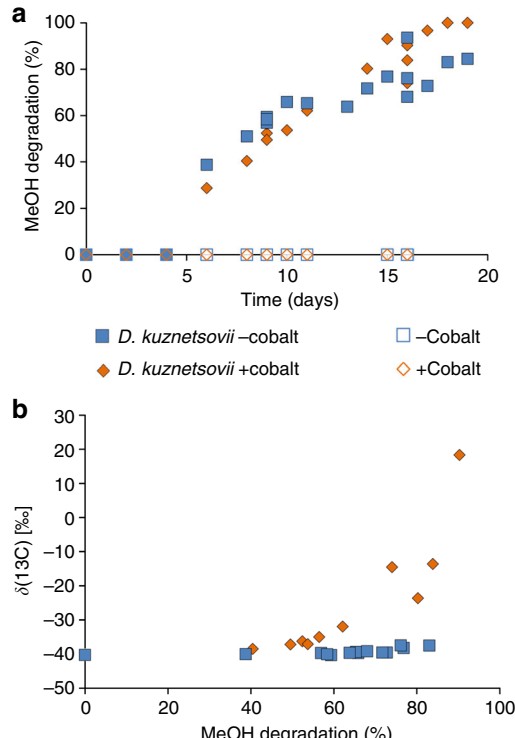

**Fig. 5** Stable carbon isotope fractionation analysis of *D. kuznetsovii*. **a** Percentage of methanol degraded in time. **b** SCIF analysis data, presented as the delta $^{13}$C fractionation values of methanol set out against the methanol degradation. Open symbols are controls without bacteria

carbon isotope fractionation necessarily. Therefore, we theorized that methanol degradation via the methyltransferase pathway in *D. kuznetsovii* will show a large isotope fractionation, while methanol degradation via the alcohol dehydrogenase pathway might result in a significantly smaller isotope effect, allowing both pathways to be differentiated by carbon stable isotope analysis. For the stable carbon isotope fractionation (SCIF) analysis cells were grown with methanol and sulfate in the presence and absence of cobalt and vitamin B12. The percentage of degraded methanol in time was measured (Fig. 5a) and delta $^{13}$C fractionation was set out against percentage of degraded methanol (Fig. 5b). A strong carbon isotope fractionation effect was observed in cobalt-amended cultures exclusively (Fig. 5b); a carbon isotope enrichment factor of $-23.8 \pm 8.6$‰ was determined and the correlation coefficient ($R^2$) of the Rayleigh plot was 0.81 (Supplementary Fig. 2). In the bottles with medium excluding cobalt and vitamin B12, no significant carbon isotope fractionation was measured during the course of methanol degradation, indicating that methanol oxidation by the ADH is indeed not associated to a carbon isotope effect.

These results show that in the medium without cobalt, the condition in which *D. kuznetsovii* predominantly synthesized the methanol-oxidizing ADH, no significantly fractionation occurs. In the medium with cobalt, the condition in which *D. kuznetsovii* also synthesized the methanol methyltransferases, considerable fractionation was observed. As can be seen in the double-logarithmic Rayleigh plot (Supplementary Fig. 2), in the medium with cobalt isotope fractionation started to occur after a certain amount of methanol was degraded. This strongly suggests that initially the ADH is involved and that the methanol methyl-transferase is operating later at lower methanol concentrations.

**Role of hydrogenases in the alcohol metabolism**. Genes coding for four hydrogenases were described to be present in the genome of *D. kuznetsovii* (Desku_0995, 2307–2309, 2934, 2995-297). All four are cytoplasmic FeFe hydrogenases. Two were suggested to be confurcating (Desku_2307–2309; 2995–2997) due to their similarity to the bifurcating/confurcating hydrogenases of *Pelotomaculum thermopropionicum*[7]. The two possible confurcating

hydrogenases were synthesized during growth of *D. kuznetsovii* with different substrates (Fig. 1), while the other two predicted hydrogenases were not identified in the proteome data. One of the confurcating hydrogenases (Desku_2307–2309) showed increased abundance during growth with methanol, whereas the other hydrogenase (Desku_2995–2997) was more abundant when *D. kuznetsovii* was grown with lactate or ethanol.

As the ADH was shown to reduce NAD[+34], the NADH and reduced ferredoxin formed by the ADH and the aldehyde ferredoxin oxidoreductase, respectively, could be used by the confurcating hydrogenase to form hydrogen. Subsequently, hydrogen could be used to reduce sulfate as proposed in a hydrogen-cycling model for sulfate reducers[44]. The abundance of the two hydrogenases was associated with the hydrogen levels that could be measured in the cultures. When grown with ethanol the hydrogen levels reached values of around 3000 ppm, while with methanol the hydrogen level was substantially lower (highest value about 550 ppm).

**Environmental implications**. The presence of two methanol-degradation pathways may be beneficial for *D. kuznetsovii* in its deep-subsurface habitat where it has to compete with other methylotrophic anaerobes. Generally, methanogens and acetogens grow faster with methanol than sulfate reducers, but their growth is hampered by cobalt limitation[35,36,45]. Methanogens appear to compete better for cobalt during cobalt-limiting conditions[36], while acetogens outcompete methanogens when the concentrations of methanol and cobalt are high[35]. Mixed culture experiments of the acetogen *Moorella thermoautotrophica* and *D. kuznetsovii* at methanol-limiting conditions showed that *D. kuznetsovii* has a higher affinity for methanol[46]. Owing to the two methanol-degrading pathways *D. kuznetsovii* can successfully compete with both methanogens and acetogens. During cobalt-limiting conditions, *D. kuznetsovii* can compete with methanogens because of the cobalt-independent pathway; and when cobalt is not limiting, but methanol concentrations are low, *D. kuznetsovii* can compete with acetogens by virtue of its methanol methyltransferase pathway.

Methanol is a common substrate in both aerobic and anaerobic environments. To analyze methanol utilizers in the environment molecular tools are required. Kolb and Stacheter addressed this issue[47]. To get a better understanding of the global methanol conversion, they discussed the need for suitable gene targets to analyze methanol-utilizing microorganisms. Moreover, they identified five potential gene markers for aerobes and one for strict anaerobes, the *mtaC* gene[47]. However, the *mtaB* gene is a better alternative as a target to develop gene-based detection of strict anaerobic methanol utilizers in the environment, because the *mtaB* codes for the methanol specific subunit of the methyltransferase. Furthermore, the MtaC has high similarity with the cobalamin binding subunits of the tri-, di-, and mono-methylamine methyltransferases. In addition to *mtaB*, another gene marker needs to be developed to target methanol-utilizing microorganisms that employ the MDH pathway as found in *D. kuznetsovii*. However, the methanol-oxidizing ADH of *D. kuznetsovii* clusters together with ADHs of species that cannot grow with methanol (Fig. 4). More methanol-degrading SRB should be investigated to assess if the use of a methanol-oxidizing alcohol dehydrogenase is more common among sulfate reducers. Moreover, finding more of these proteins will help establishing the difference with only ethanol-oxidizing ADHs and will lead to a suitable gene marker.

**New hypothetical energy-conserving formate dehydrogenase complex**. Growth of *D. kuznetsovii* on lactate led to increased abundances of lactate transporter and lactate dehydrogenase (Desku_2393–2995), pyruvate formate lyase (Desku_2520) and likely a formate dehydrogenase complex (Desku_0187–0192). The use of a pyruvate formate lyase instead of ferredoxin-oxoacid (pyruvate ferredoxin oxidoreductase (Deku_0030–0033), which is not more abundant when grown with lactate, might be beneficial from the energetic point of view. Research with *D. vulgaris* indicated that intracellular cycling of formate formed by pyruvate formate lyase might contribute to energy conservation[48]. Formate conversion to hydrogen and carbon dioxide indeed is coupled to energy conservation and growth of *Desulfovibrio*, even in the absence of sulfate[49,50]. Interestingly, the presumed formate dehydrogenase complex of *D. kuznetsovii* does not have much similarity with any of the formate dehydrogenases of *D. vulgaris*. The formate dehydrogenase complexes (Desku_0187–0192 and Desku_2987–2991) need to be studied further. The abundant protein complex when grown with lactate, Desku_0187–0192, consists of five subunits. Desku_0187 and 0188 are annotated as a glutamate synthase and a FAD dependent oxidoreductase, respectively. Both protein sequences contain several pyridine nucleotide-disulfide oxidoreductase domains, which indicates that these subunits are the catalytic subunits of the protein complex. The annotated glutamate synthase (Desku_0187) has about 62 % similarity with a FAD nucleotide disulfide oxidor-eductase of *Desulfotomaculum ruminis*. The Desku_0189 is annotated as a methylviologen-dependent hydrogenase. The protein annotated as a formate dehydrogenase (Desku_0190) contains a 4Fe-4S dicluster, but lacks the characteristic catalytic domain of other formate dehydrogenases. Therefore, we hypo-thesize that this enzyme complex concerns a novel type of energy-conserving formate dehydrogenase complex. The protein sequence of Desku_0192 predicts a Twin-arginine signal peptide cleavage site, but none of the subunit sequences of the complex predicts transmembrane helixes. This indicates that the membrane complex is translocated across the membrane. Currently, it is unclear if Desku_0184–0186 also belong to the enzyme complex. Desku_0185 is also more abundant when grown with lactate, while Desku_0184 and Desku_0186 are not found in the proteome.

## Methods

**Culture medium and experimental design**. *Desulfotomaculum kuznetsovii*[10] was grown in bicarbonate buffered medium described by Stams et al.[51] To investigate whether a methanol methyltransferase system is involved in methanol conversion, *D. kuznetsovii* was grown with methanol and sulfate in normal medium (i.e., using the trace and vitamin solutions described by Stams et al.[51], containing $CoCl_2$ and vitamin B12) and in medium deprived from cobalt ($CoCl_2$) and vitamin B12. Methanol (20 and 5 mM) and sulfate (10 mM) were added from concentrated stock solutions (sterilized by autoclaving). In addition to the four methanol growth conditions (20 mM methanol with and without cobalt and vitamin B12, and 5 mM methanol with and without cobalt and vitamin B12), two other growth conditions were used for a comparative proteomics analysis. Those growth conditions were: lactate (20 mM) with sulfate (10 mM) and ethanol (20 mM) with sulfate (10 mM) (both in medium containing cobalt and vitamin B12).

Cultivation of *D. kuznetsovii* was performed at pH 7 and 60 °C in 117 mL glass serum bottles with butyl rubber stoppers and aluminum crimp seals. The bottles contained 50 mL basal medium and a gas phase of 1.7 bar $N_2/CO_2$ (80%/20%, vol/vol). In initial growth experiments and the stable isotope fractionation experiment the inoculum size was 1% (vol/vol) and cultures were transferred at least five times to ensure full adaptation to the growth substrate. For proteomics, cultures were transferred at least ten times. Assays for proteomics were performed in triplicate or quadruplicate.

Growth was recorded by monitoring the optical density at 600 nm (U-1500 spectrophotometer Hitachi), by gas chromatographic determination of the methanol concentration (using a GC-2010, Shimadzu, equipped with a Sil 5 CB column), and by measuring sulfate concentrations using ion-chromatography (an ICS2100 system, Thermo Scientific, equipped with an AS19 column). Sulfide was measured photometrically with the methylene blue method[52]. Hydrogen in bottles' headspace was monitored by gas chromatography (using a Compact GC4.0, Global Analyser Solutions, equipped with Carbonex 1010 column (Supelco, 3 m × 0.32

mm) followed by a Mosieve 5A column (Restek, 30 m × 0.32 mm) and a thermal conductivity detector (TCD)).

**Protein extraction**. For the preparation of protein samples, all six conditions of 250 mL cell suspensions, including their independent replicates, were grown and cells harvested by centrifugation when ~70–80% of the substrate was depleted. The pellets were resuspended separately in SDT-lysis buffer (100 mM Tris/HCl pH 7.6 +4 % SDS, vol/vol+0.1 M dithiothreitol) and sonicated (Sonifier B12, Branson Sonic Power Company, Danbury, CT) to trigger disruption of the bacterial cell wall. Unbroken cells and debris were removed by centrifugation at 15,700×g for 10 min. The protein containing supernatant was used for the proteome analysis.

**Comparative proteomics**. The proteome analysis of *D. kuznetsovii* cells grown in the six growth conditions were performed using nanoLC-MS/MS. Overall, 40 µg of protein was separated by SDS-PAGE on a 10-well SDS-PAGE 10% (wt/vol) Bis-Tris Gel (Mini Protean System, Bio-Rad, San Diego, CA), for 55 min at a constant voltage of 120 V using Tris-SDS as running buffer. Gels were stained with Colloidal Blue Staining Kit (Life Technologies, Carlsbad, CA) and treated for reduction and alkylation using 10 mM dithiothreitol and 15 mM iodoacetamide in 50 mM ammonium bicarbonate. Each lane was cut into 4 even slices and each slice was cut into small pieces of ca. 1–2 mm². Digestion was performed by adding 50 µL of sequencing grade trypsin (5 ng/µL in 50 mM ammonium bicarbonate) and incubated at room temperature overnight while shaking. The resulting tryptic peptide samples were desalted and subjected to nanoLC-MS/MS using a Proxeon Easy nanoLC and an LTQ-Orbitrap XL instrument (Thermo Fisher Scientific, Naarden, the Netherlands) as described earlier[53].

LCMS runs with all MSMS spectra obtained were analyzed with MaxQuant 1.5.2.8[54] using the "Specific Trypsin/P" Digestion mode with maximally two missed cleavages and further default settings for the Andromeda search engine (First search 20– ppm peptide tolerance, main search 4.5 ppm tolerance, ITMSMS fragment match tolerance of 0.5 Da, Carbamidomethyl (C) set as a fixed modification, while variable modifications were set for protein N-terminal acetylation and M oxidation, which were completed by non-default settings for de-amidation of N and Q[55].

A *D. kuznetsovii* database downloaded from Uniprot (http://www.uniprot.org) 16 May 2017, containing 3387 entries was used together with a database containing most common external protein contaminants. The "label-free quantification" as well as the "match between runs" options were enabled. De-amidated peptides were allowed to be used for protein quantification and all other quantification settings were kept default.

Filtering and further bioinformatic analysis of the MaxQuant/Andromeda workflow output and the analysis of the abundances of the identified proteins were performed with the Perseus 1.5.5.3 module (available at the MaxQuant suite). Accepted were both peptides and proteins with a false discovery rate (FDR) of less than 1% and proteins with at least two identified peptides of which at least one should be unique and at least one should be unmodified. Reversed hits were deleted from the MaxQuant result table as well as all results showing a normalized label-free quantitation intensity (LFQ) value of 0 for both sample and control. From the original 1622 protein groups in the original MaxQuant result, 208 were filtered out leaving 1414 protein groups. The logarithm (base 10) was taken from protein LFQ MS1 intensities as obtained from MaxQuant. Relative protein quantitation of sample to control was done with Perseus by applying a two sample *t*-test using the "LFQ intensity" columns obtained with FDR set to 0.05 and S0 set to 1. Total non-normalized protein intensities corrected for the number of measurable tryptic peptides (intensity based absolute quantitation (iBAQ) intensity[56] were, after taking the normal logarithm, used for plotting on the y-axis in a Protein ratio vs. abundance plot. nanoLC-MSMS system quality was checked with PTXQC[57] using the MaxQuant result files.

**Stable isotope fractionation analysis**. For the stable isotope fractionation analysis, 31 bottles were prepared with normal medium (described above) and 31 bottles were prepared that contained medium without cobalt and vitamin B12. To increase the sensitivity of isotope fractionation analysis (see below), the methanol concentration was increased to 40 mM. This concentration is not toxic to *D. kuznetsovii*, but growth stopped after degrading ~25 mM of methanol (data not shown), resulting from sulfate reduction to sulfide, which reached growth-inhibitory concentrations. To prevent increasing concentrations of sulfide during growth, iron(II) was included in the medium to react with the sulfide and precipitate. Each bottle contained 40 mM methanol and 30 mM iron(II) sulfate. Twenty-one bottles of each medium were inoculated with 1 % (v/v) active *D. kuznetsovii* and ten bottles of each medium served as non-inoculated controls. All bottles were incubated at 60 °C.

At different time points a bottle of each medium was inactivated by adding concentrated sodium hydroxide to a pH above 12. Samples were taken to monitor methanol and sulfate concentrations before adding sodium hydroxide to a bottle. Moreover, sodium sulfide was also added before adding sodium hydroxide to precipitate all iron(II) from the medium. After inactivation bottles were stored at 4 °C.

Prior to the stable isotope fractionation analysis calcium chloride was added to precipitate the carbonate from the medium and the medium was centrifuged (MiniSpin®, Eppendorf, Hamburg, Germany) for 5 min at 12,000×g and room temperature to remove the carbonate and iron precipitates.

High performance liquid chromatography coupled via LC-IsoLink interface to a stable isotope ratio mass spectrometer MAT 253 (Thermo Fisher Scientific, Bremen, Germany) was used to determine the carbon stable isotope ratios of methanol following the principle of a wet chemical oxidation as described before[58]. The HPLC system was further equipped with a HTC PAL autosampler (CTC Analytics, Zwingen, Switzerland), a Surveyor MS Pump Plus (Thermo Fisher Scientific, Bremen, Germany), and a HT HPLC 200 column oven (SiM, Oberhausen, Germany). Sample aliquots (10 µL) were separated on an Atlantis T3 column (150 mm × 3 mm, 3 µm inner diameter; Waters, Eschborn, Germany), equipped with a 10 mm × 2.1 mm pre-column (Waters, Eschborn, Germany) at 40 °C using Milli-Q water with a flow rate of 100 µL/min as eluent. The wet chemical oxidation of methanol was achieved by online mixing with ortho-phosphoric acid (0.75 M) and sodium peroxodisulfate (200 g/L) prior to entering the oxidation reactor. The reagents were pumped separately by two pumps with flow rate of 50 µL/min each. The temperature of the reactor was maintained at 99.9 °C. All samples were measured in triplicate, and the typical uncertainty of analysis was < 0.4 ‰. Enrichment factors and standard deviations were calculated as described in Jaekel et al.[59]. The error of the enrichment factor is given as 95% confidence interval (CI), determined using a regression analysis as described by Elsner et al.[60]

**Data availability**. The mass spectrometry proteomics data have been deposited to the ProteomeXchange Consortium via the PRIDE[61] partner repository with the data set identifier PXD006899. All other relevant data are available in this article and its Supplementary Information files, or from the corresponding author upon request.

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

## Acknowledgements

Research was funded by grants of the Division of Chemical Sciences (CW-TOP 700.55.343) and Earth and Life Sciences (ALW 819.02.014) of The Netherlands Organisation for Scientific Research (NWO), the European Research Council (ERC grant 323009), and the Gravitation grant (024.002.002) of the Netherlands Ministry of Education, Culture and Science.

## Author contributions

D.Z.S., M.V. and A.H.V.G. performed physiological and proteomics experiments and analyzed the data. S.B., M.M.P., M.W.H.P., P.D.E.M.V. did peptide analyses and analyzed proteomics data. C.V., S.F. and S.K. performed stable isotope fractionation analysis and

contributed with the data analysis. M.V., D.Z.S. and A.J.M.S. designed the experiments and wrote the manuscript. All authors agreed with the final version of the manuscript.

## Additional information

**Competing interests:** The authors declare no competing financial interests.

