## [Peer Review File · Nature Communications]

Reviewer #1 (Remarks to the Author):

The manuscript by Visser et al. examines the methanol metabolism of *D. kuznetsovii* using a proteomic approach reconciled with its annotated genome. Experimentally the cells were grown either with cobalt plus vitamins or without. Evidence was uncovered for two basic pathways, one involving alcohol dehydrogenases which also acted as methanol dehydrogenases in a manner analogous to aerobic methanotrophs, sequentially oxidizing methanol to formate and thence to CO₂. It is found in anaerobic acetogens. That pathway did not require Co. The other pathway was analogous to that of methanogens, required Co and had methyl transferases and folates as carriers on the way to CO₂. In both cases, electron generated were shunted to reduce sulfate to sulfide. The interpretation was further supported by the stable C isotopic fractionation effort which exhibited strong fractionation at low methanol concentrations (when the folate pathway was working) and low fractionation at high methanol concentrations when the alcohol dehydrogenase was operative. Hence, *D. kuznetsovii* has two pathways that allow it to compete with methanogens and acetogens for methanol in the deep subsurface.

I can't find anything wrong with this really very nice piece of work, other than on line 79 the authors made the grievous mistake of miss-typing the word "importantly." Since that error can be readily fixed, the work is essentially flawless.

Reviewer #2 (Remarks to the Author):

The manuscript authored by Dr. Visser and colleagues explores the methanol oxidation of *Desulfotomaculum kuznetsovii*. The genome sequence annotation showed genes that supported the possibility of two pathways for the degradation of methanol, one that required a corrinoid protein and an alternative path that was alcohol dehydrogenase dependent. Additional evidence for the two pathways was obtained from a proteomics analysis of sulfate-respiring cells growing on methanol, with or without cobalt and B12, compared with ethanol or lactate as electron and carbon donors.

The stable isotope fractionation analysis showed a major difference between the two pathways and confirmed the operation and conditions of operation of the alternative enzymes. These data were incontestable.

This is an interesting research story with several lines of support. However, there is one serious question that will need to be clarified. The proteomics data illustrated in Figure 1 would appear to be analyzed problematically. In the legend of that figure the last sentence states, “Values range between 0 and 1 and are calculated by comparing peptide abundance of one protein of a condition to total peptide count of all conditions.” This would mean that the expression/observation of protein in one condition will affect the interpretation of expression in other conditions. However, these are independent experiments and there should be no interdependence of the results.

A sum of one protein in all conditions would vary when only one growth condition differed in protein content. If 10 peptide observations were made in each of the four conditions, the math would indicate a value of 0.25 (10/40) for each growth condition. If the protein failed to be found in one condition but 10 peptides were seen in the other three, now the value for expression in one of the three would be 0.33 (10/30). One might think there was an increase in expression but that would not be true. The actual peptide counts might be less confusing.

When discussing the hydrogenases, it would have been useful to have measured hydrogen in the culture as supporting the interpretation. Interestingly, lines 163-164 are written as if the “hydrogen-cycling model” was an established fact. You might want to add a reference where that has been proved.

A number of editorial changes should be considered.

- 1) L 27: “Sulfate-respiring cells grown with methanol... [line 28]...with cells grown with lactate or ethanol.”
- 2) L.79: Importantly—spelling.
- 3) L. 90: Was the lower rate of methanol oxidation by ADH actually statistically significant? The data shown for methanol growth in Fig. 5 showed that the growth in the absence of cobalt was slower only after 10 days (~70% of methanol used). That is about the time that the isotope fractionation begins in earnest.

- 4) L. 118-9: When you state that the abundance of proteins is higher or lower in one condition versus another, please put the peptide counts in parentheses. These would support your arguments better than a reference to Fig. 1. This comment also applies to line 125, to lines 132-3, and to lines 158-159.
- 5) L. 128: Spelling of nicotinamide and dinucleotide.
- 6) L. 131: Spelling of Goorissen?
- 7) L. 147: “which is a remarkable, unexpected finding.” or “which is a remarkably unexpected finding.” The choice depends on what you meant to say. When discussing this finding, would you not consider horizontal gene transfer as the source of the enzyme?
- 8) L. 160: “...grown with lactate or ethanol.”
- 9) L. 176-7: “...in a significantly smaller isotope... allowing both pathways to be differentiated by carbon stable isotope analysis.”
- 10) Starting L. 195...This is a nicely argued implication!
- 11) L. 225-227: You have two genera both referred to as *D*. Perhaps they should be *Dp*. and *Dm*.
- 12) L. 232: elements ?
- 13) Throughout the M&Ms, please remember that when % is used as a concentration term, it must be defined as either wt/wt, wt/vol, or vol/vol.
- 14) Also remember Gram is a surname.
- 15) L. 262: “...were performed to identify...”
- 16) L. 264: “...and subjected to electrophoresis for 5 min...”
- 17) L. 266: Spelling dithiothreitol.
- 18) L. 277: Define HCD.
- 19) L. 300: precipitate not precipitates.
- 20) L. 303: “...a bottle of each medium was inactivated...”
- 21) L. 308: Please change the centrifugation conditions to xg.
- 22) Ref. 11: Is this journal correct?
- 23) Ref. 33, please add year.

Reviewer #3 (Remarks to the Author):

This manuscript reports analyses performed on pure cultures of *D. kuznetsovii* grown with lactate, methanol, methanol-cobalt-B12, or ethanol as the energy source. Cells were collected from each condition and used for label-free quantitative proteomics to observe differentially expressed proteins. In parallel experiments, the isotopic fractionation of methanol was quantified in cultures grown on methanol and methanol-cobalt-B12. The results indicate that proteins corresponding to two distinct pathways are expressed during growth on methanol, one for a cobalamin dependent methyltransferase system related to that in methylotrophic archaea and one involving a methanol dehydrogenase and aldehyde:Fd oxidoreductase. The existence of two distinct methanol oxidation pathways was supported by a difference in isotopic fractionation of methanol depending on the cobalt/B12 status or methanol concentration of the culture. The conclusion based on correlation of these observations is that *D. kuznetsovii* employs two distinct methanol oxidation pathways depending on prevailing conditions. I generally agree with the conclusions and agree with the arguments of the authors that this would be a fairly novel finding. It is tempered by the fact that there are multiple methanol dehydrogenases known in aerobic methylotrophic bacteria, i.e PQQ-dependent and lanthanide dependent. One could argue these are "two pathways", but these are still both just alcohol dehydrogenases, one of which uses fairly exotic metals for cofactor. It would help to solidify the point here that these are two drastically different pathways by citing and briefly discussing the literature on MDH for comparison. This would help with broader context.

The change in conversion rate between conditions in Fig. S1 is not very convincing. What is the actual difference in linear regression between days 3-10? I don't think the rate will differ by more than 5-10%. Was there a difference in growth rate or cell yield between conditions or were these substrate turnovers measured under non-growth conditions? How much of a difference in growth rate might you expect based on this difference in conversion? The two pathways indicate to me that cell yield should be somewhat higher when the MT system is employed since intermediates in the C1-THF pool are available for assimilation into biomass whereas the carbon in the ADH pathway should be predominantly lost or at least only assimilated after oxidation to CO₂. It would probably still be a subtle difference, but it would be good to discuss this data if available. Is there any evidence in the genome of a formaldehyde assimilation pathway that might help to mitigate potential differences in yield between the two proposed pathways?

What is the evidence that Desku_2307-9 hydrogenase is confurcating? This argument needs to be better established or the mechanism in Fig. 2 does not necessarily work because the cells would not be able to recycle either the NAD pool or Fd pool to continue growth. Similarly, what is the electron carrier and proposed mechanism for energy conservation when the cells are utilizing the Mta pathway? This is not clear and is potentially related to the point in the previous paragraph. Is the energy conservation expected to be more efficient by not requiring H₂ as an intermediate as suggested by this figure? Addressing this issue will help make the manuscript more accessible to a broader audience. Also, at physiological pH, the relevant species for the terminal reduction product is most likely HS⁻.

Fig. 5A does not add too much, it could be omitted and simply mentioned in the text that the extent of substrate oxidation in the two conditions was similar or moved to supplemental. For Fig. 5B, on what substance is the SCIF being measured? Biomass, methanol? It is not clear from the text at this point and the reader has to wade down to the methods to dig this out. Please specify here. I don't necessarily agree with the "strongly" in l. 192 because the methanol concentration in the SCIF experiment was very high. Could it be that the the ADH pathway is primarily for rapid detoxification and happens to help when cobalt is not present? How realistic in 40 mM methanol in an environment where it is being abiotically produced from CO₂ and H₂? Was methanol detected in the geothermal reservoir where *D. kuznetsovii* was isolated from? If so, what concentration?

The environmental implications suggest that these *D. kuznetsovii* can compete with poorer cobalt scavenger *Moorella thermoacetica*, but what about with a methanogen that the authors suggest

should be even better than *M. thermoacetica*? This would suggest that *D. kuznetsovii* would really "win" only in a very low cobalt, but high methanol environment. Do such environments exist in nature? Cobalt can exist as cobalt sulfide, though the melting point is quite low. Does driving a system to high sulfide content deprive every microbe in the system of cobalt and could *D. kuznetsovii* potentially end up creating this niche by its own metabolism?

One of the questions to consider in the review is "...do you feel that the paper will influence thinking in the field?" This is difficult. *D. kuznetsovii* will set a precedent, but methanol oxidizing sulfate reducing bacteria are not necessarily on every microbiologist's radar. Many will consider them "boutique" and not broadly applicable. So, in the field of C1 metabolism, this will have an effect, but I am uncertain about broader audiences. I think the argument can be made, but was not made effectively in this version.

Minor

There are a number of awkward sentences, phrasings, and misspellings. I have pointed out a few here, but I suggest that the editor help the authors to clean this up before any publication.

I. 46 Why is "extremely" in parentheses? Are these significantly more heat resistant than a garden variety endospore from *Bacillus* spp.?

I. 79 Importnatly -> Importantly

II. 143-144 "divides two big clusters" is awkward. Perhaps "displays two major clades"

Fig. 3 Why not use the locus tag in this figure for the *D. kuznetsovii* protein as in other figures?

I. 177 allowing to differentiate both pathways -> allowing the pathways to be differentiated

II. 259-260 Without staining, imaging, and comparison, I'm not sure how this serves as a quality control. What parameters or qualities were used to judge if a sample was of sufficient quality to proceed?

I. 261 120 mV? Did you mean V here?

II. 263-265 What is the purpose of this step? Why not proceed directly from the liquid sample?

I. 273 What was the electrospray voltage?

I. 279 ThermoFischer->Thermo Fisher ?

Reviewer #4 (Remarks to the Author):

The manuscript proposed by Visser, Pieterse, Pinkse, and co-authors describes the proteomic analysis of *Desulfotomaculum kuznetsovii*, isolated from a 3000 m deep geothermal water reservoir and described in 1988. The genome analysis of *D. kuznetsovii* strain 17(T) was published in 2013 by the same research team. Here, the authors focused their attention on the key players involved in methanol metabolism by comparing cells grown with methanol and cells grown with other substrates. Their results show that two methanol degrading pathways are used if considering induced pathways. Furthermore, isotope fractionation allows deciphering the role of each of these pathways at low and high methanol concentrations. I found the manuscript mainly descriptive as it relies only on proteomics and isotope mass spectrometry experiments, and phylogenetic analysis. As per the editorial request, I am commenting on the proteomic analyses included in this report. The proteomic conclusions are based on only two biological replicates per condition and semi-quantitation of proteins by the spectral count approach. The raw data are not available and it is difficult to judge the statistics of these datasets. Central metabolism is modulated quite drastically in bacteria and the authors could easily point at the key proteins involved in methanol metabolism. In this sense, I am expecting a more in-depth characterization of the corresponding enzymes than just a phylogenetic tree.

1. The comparative proteomic data analysis comprises four different conditions but only two

biological replicates. This gives a rather low confidence in these results. Furthermore, the authors mentioned (line 96) that the lactate growth condition was used as a reference. Table S2 shows that this reference is not of quality, with 6469 MS/MS spectra assigned for the first replicate and 8624 MS/MS spectra assigned for the second replicate. The second replicate dataset is thus inflated of +33%! Usually, a normalization of the protein material should be done for comparing equal quantities of material. Here, the reference shows unusual variations. I recommend to the authors to consider new biological replicates and precise normalization of their samples before MS/MS recording.

2. Another problem appears when the different conditions are compared. In the MeOH -B12 condition (replicates 1 and 2), more MS/MS spectra are recorded (in average +17%) compared to the MeOH condition (replicates 1 and 2). The authors should normalize the protein quantities before peptide extraction and recording their tandem mass spectrometry datasets.

3. Protein abundances have been quantified based on MS/MS spectral count and not peptide count. The authors should modify their manuscript accordingly. In addition, such measure is valid if the tandem mass spectrometer has enough time to record several spectra per peptides if abundant. The authors performed a 60 min gradient with a dynamic exclusion time set to 30 seconds. This gradient is rather short for a comprehensive view of a proteomic sample, but I expect that a dynamic exclusion of no more than 10 sec is activated for spectral count. Therefore, only the most abundant proteins will be quantified with enough spectral counts for comparative proteomics. I found these settings not relevant here.

4. Table S2 shows only a partial view on the proteins identified. Some ambiguities are noted: i) Proteins that are 3 times more abundant in one condition as indicated in the manuscript, or proteins with at least 4 peptide counts as mentioned in Table S2? Table S2 lists 626 proteins while 892 have been identified with at least two different peptides. Selecting and presenting only partial proteomic data is really unusual. I wonder what's the reason...

5. Another point is the growth of the bacteria. The authors mentioned that cells were harvested at the late exponential phase. How this growth was followed? Are the authors sure that the same growth stage have been compared? Is this based on a simple optical density measurement, or a sound kinetic analysis with growth curve and equivalent growth stages?

6. As recommended by the proteomics community, the raw data should be deposited in a public repository where reviewers may have access for checking their quality.

7. Figure 1 shows a heatmap of selected proteins. The authors did not use the lactate reference, but rather proposed a peptide count of all conditions for calculating the fold changes. This is rather unusual. The statistics to validate these fold changes are not presented and are not explained. A p value for each protein fold change should be available, as well as a Benjamin-Hochberg post-hoc test.

8. Some predicted functions are surprisingly different between Table S2 and Figure 1 (see for example, the Desku_1072 and Desku_0050 protein labels).

9. The analysis of the protein dataset is superficial. For example, Desku_0057 is annotated as a ferredoxin and is highly abundant (as much as MtaA). In fact, a simple BLAST analysis shows that this protein is not a ferredoxin but rather a multi-domain protein including an Fe-S ferredoxin-like domain at its N-terminal.

10. The study could be complemented with comparative genomics of the whole pathways, rather than phylogenetic trees.

11. The authors cited a reference from 2002 (ref 51) to support their LC-MS/MS analysis. This reference has nothing related to the instrument used here, a Q-Exactive Plus tandem mass spectrometer.

12. The general quality of the figures and the supplementary material is low. How these data are normalized? Where are the fold changes and associated p values for comparing these results?

13. I wonder if it is the strain 17(T) that has been analyzed by proteomics? This is not mentioned in the manuscript.

We would like to thank all reviewers for their feedback.

Answers to Reviewer #1:

The manuscript by Visser et al. examines the methanol metabolism of *D. kuznetsovii* using a proteomic approach reconciled with its annotated genome. Experimentally the cells were grown either with cobalt plus vitamins or without. Evidence was uncovered for two basic pathways, one involving alcohol dehydrogenases which also acted as methanol dehydrogenases in a manner analogous to aerobic methanotrophs, sequentially oxidizing methanol to formate and thence to CO₂. It is found in anaerobic acetogens. That pathway did not require Co. The other pathway was analogous to that of methanogens, required Co and had methyl transferases and folates as carriers on the way to CO₂. In both cases, electron generated were shunted to reduce sulfate to sulfide. The interpretation was further supported by the stable C isotopic fractionation effort which exhibited strong fractionation at low methanol concentrations (when the folate pathway was working) and low fractionation at high methanol concentrations when the alcohol dehydrogenase was operative. Hence, *D. kuznetsovii* has two pathways that allow it to compete with methanogens and acetogens for methanol in the deep subsurface.

I can't find anything wrong with this really very nice piece of work, other than on line 79 the authors made the grievous mistake of miss-typing the word "importantly." Since that error can be readily fixed, the work is essentially flawless.

Thank you for the positive feedback.

Answers to Reviewer #2:

The manuscript authored by Dr. Visser and colleagues explores the methanol oxidation of *Desulfotomaculum kuznetsovii*. The genome sequence annotation showed genes that supported the possibility of two pathways for the degradation of methanol, one that required a corrinoid protein and an alternative path that was alcohol dehydrogenase dependent. Additional evidence for the two pathways was obtained from a proteomics analysis of sulfate-respiring cells growing on methanol, with or without cobalt and B12, compared with ethanol or lactate as electron and carbon donors. The stable isotope fractionation analysis showed a major difference between the two pathways and confirmed the operation and conditions of operation of the alternative enzymes. These data were incontestable. This is an interesting research story with several lines of support. However, there is one serious question that will need to be clarified. The proteomics data illustrated in Figure 1 would appear to be analyzed problematically. In the legend of that figure the last sentence states, "Values range between 0 and 1 and are calculated by comparing peptide abundance of one protein of a condition to total peptide count of all conditions." This would mean that the expression/observation of protein in one condition will affect the interpretation of expression in other conditions. However, these are independent experiments and there should be no interdependence of the results. A sum of one protein in all conditions would vary when only one growth condition differed in protein content. If 10 peptide observations were made in each of the four conditions, the math would indicate a value of 0.25 (10/40) for each growth condition. If the protein failed to be found in one condition but 10 peptides were seen in the other three, now the value for expression in one of the three would be 0.33 (10/30). One might think there was an increase in expression but that would not be true. The actual peptide counts might be less confusing.

We understand the reviewer's concern regarding Figure 1. As we had repeated the proteome experiment and the data were evaluated differently, the concerns of the reviewers are not applicable any more.

When discussing the hydrogenases, it would have been useful to have measured hydrogen in the culture as supporting the interpretation. Interestingly, lines 163-164 are written as if the "hydrogen-cycling model" was an established fact. You might want to add a reference where that has been proved.

We agree that the hydrogen-cycling model has not been fully proven or refuted. The text in the manuscript was rephrased (L 206-207). We did measure hydrogen production in cultures growing on methanol and ethanol (see table below). With methanol the hydrogen levels are substantially lower than with ethanol (for similar OD), which might indicate the involvement of one of the two different hydrogenases with either substrate.

Bottle	Co/B12 + or -	Carbon source	4 days incubation	6 days incubation		10 days incubation	
			H ₂ (ppm)	OD (600 nm)	H ₂ (ppm)	OD (600 nm)	H ₂ (ppm)
1	-	MeOH20	18	0.08	215	0.11	520
2	-	MeOH20	20	0.07	218	0.08	555
3	-	EtOH20	1999	0.22	3075	0.19	2337
4	-	EtOH20	1468	0.28	3310	0.25	3283
5	+	MeOH20	0	0.07	37	0.25	330
6	+	MeOH20	0	0.07	26	0.20	367
7	+	EtOH20	911	0.20	1558	0.18	1382
8	+	EtOH20	1899	0.25	1780	0.24	1502

Note: Assays were performed with 20 mM carbon source and 15 mM sodium sulfate.

A number of editorial changes should be considered.

- 1) L 27: "Sulfate-respiring cells grown with methanol... [line 28]...with cells grown with lactate or ethanol." *Corrected.*
- 2) L.79: Importantly—spelling. *Corrected.*
- 3) L. 90: Was the lower rate of methanol oxidation by ADH actually statistically significant? The data shown for methanol growth in Fig. 5 showed that the growth in the absence of cobalt was slower only after 10 days (~70% of methanol used). That is about the time that the isotope fractionation begins in earnest.

The reviewer raised an important detail here. We calculated methanol conversion rates using a non-linear fitting of the data for each of the triplicate assays; after t-test analysis we concluded that methanol conversion rates in the presence or absence of ethanol are not statistically significant. What is relevant in fig. S1 is the fact that methanol levels stabilize at statistically significant ($p=0.00027$) lower concentrations in assays with the addition of cobalt (final average concentration of residual methanol in assays with and without cobalt is 0.82 ± 0.53 mM and 2.68 ± 0.65 mM, respectively). This suggests the importance of methyltransferase system for the conversion of low methanol concentrations. Text in the manuscript was changed L 97-101.

	+ cobalt	- cobalt
methanol conversion rate (mmol L ⁻¹ day ⁻¹)	2.63±0.49	2.53±0.06
residual methanol concentration (mM)	0.82±0.53	2.68±0.65

- 4) L. 118-9: When you state that the abundance of proteins is higher or lower in one condition versus another, please put the peptide counts in parentheses. These would support your arguments better than a reference to Fig. 1. This comment also applies to line 125, to lines 132-3, and to lines 158-159.
The concern is taken away by the new proteome experiment.
- 5) L. 128: Spelling of nicotinamide and dinucleotide. *Corrected.*
- 6) L. 131: Spelling of Goorissen? *Corrected.*
- 7) L. 147: “which is a remarkable, unexpected finding.” or “which is a remarkably unexpected finding.” The choice depends on what you meant to say. When discussing this finding, would you not consider horizontal gene transfer as the source of the enzyme?
Horizontal gene transfer is a plausible hypothesis to explain the presence of a MTA pathway in D. kuznetsovii. We added this to the manuscript, L 158-159.
- 8) L. 160: “...grown with lactate or ethanol.” *Corrected.*
- 9) L. 176-7: “...in a significantly smaller isotope... allowing both pathways to be differentiated by carbon stable isotope analysis.” *Corrected.*
- 10) Starting L. 195...This is a nicely argued implication!
- 11) L. 225-227: You have two genera both referred to as D. Perhaps they should be Dp. and Dm.
Desulfosporosinus was abbreviated just once, therefore we omitted the use of abbreviation for this genus avoiding confusion.
- 12) L. 232: elements ? *Corrected to “acid trace elements solution”.*
- 13) Throughout the M&Ms, please remember that when % is used as a concentration term, it must be defined as either wt/wt, wt/vol, or vol/vol. *This information was included for % values.*
- 14) Also remember Gram is a surname. *Corrected.*
- 15) L. 262: “...were performed to identify...” *Corrected.*
- 16) L. 264: “...and subjected to electrophoresis for 5 min...” *Corrected.*
- 17) L. 266: Spelling dithiothreitol. *Corrected.*
- 18) L. 277: Define HCD. *Defined, higher-energy collisional dissociation.*
- 19) L. 300: precipitate not precipitates. *Corrected.*
- 20) L. 303: “...a bottle of each medium was inactivated...” *Corrected.*
- 21) L. 308: Please change the centrifugation conditions to xg. *Corrected.*
- 22) Ref. 11: Is this journal correct? *We checked the Journal again and it is correct.*
- 23) Ref. 33, please add year. *Publication year was added.*

Answers to Reviewer #3:

This manuscript reports analyses performed on pure cultures of *D. kuznetsovii* grown with lactate, methanol, methanol -cobalt -B12, or ethanol as the energy source. Cells were collected from each condition and used for label-free quantitative proteomics to observe differentially expressed proteins. In parallel experiments, the isotopic fractionation of methanol was quantified in cultures grown on methanol and methanol -cobalt -B12. The results indicate that proteins corresponding to two distinct pathways are expressed during growth on methanol, one for a cobalamin dependent methyltransferase system related to that in methylotrophic archaea and one involving a methanol dehydrogenase and aldehyde:Fd oxidoreductase. The existence of two distinct methanol oxidation pathways was supported by a difference in isotopic fractionation of methanol depending on the cobalt/B12 status or methanol concentration of the culture. The conclusion based on correlation of these observations is that

D. kuznetsovii employs two distinct methanol oxidation pathways depending on prevailing conditions. I generally agree with the conclusions and agree with the arguments of the authors that this would be a fairly novel finding. It is tempered by the fact that there are multiple methanol dehydrogenases known in aerobic methylotrophic bacteria, i.e PQQ-dependent and lanthanide dependent. One could argue these are "two pathways", but these are still both just alcohol dehydrogenases, one of which uses fairly exotic metals for cofactor. It would help to solidify the point here that these are two drastically different pathways by citing and briefly discussing the literature on MDH for comparison. This would help with broader context.

We agree with the reviewer that it has to be clearly stated in the text that the two methanol-degrading pathways in D. kuznetsovii are two completely different mechanisms for methanol utilization. Therefore, we added information, in the introduction, regarding two PQQ-MDHs in Methylobacterium extorquens and we added "distinct" in L 62-65 and L 83, respectively.

The change in conversion rate between conditions in Fig. S1 is not very convincing. What is the actual difference in linear regression between days 3-10? I don't think the rate will differ by more than 5-10%. Was there a difference in growth rate or cell yield between conditions or were these substrate turnovers measured under non-growth conditions? How much of a difference in growth rate might you expect based on this difference in conversion? The two pathways indicate to me that cell yield should be somewhat higher when the MT system is employed since intermediates in the C1-THF pool are available for assimilation into biomass whereas the carbon in the ADH pathway should be predominantly lost or at least only assimilated after oxidation to CO₂. It would probably still be a subtle difference, but it would be good to discuss this data if available.

Please see answers to Reviewer 2. There is no significant difference between methanol conversion rates with or without cobalt and vitamin B12. This information was corrected in the manuscript L 97-101.

The substrate turnovers were measured during growth. However, there are no differences in cell yield. The maximum OD values are on average 0.2 for both conditions. We added our OD measurements to supplementary data S1.

Is there any evidence in the genome of a formaldehyde assimilation pathway that might help to mitigate potential differences in yield between the two proposed pathways?

We have checked the genome for genes coding for proteins involved in formaldehyde assimilation but cannot find any. However, we cannot assume that the methanol that is degraded via the MT pathway yields more biomass. It is true that the methyl-group originating from the MT pathway enters the acetyl-CoA pathway at the methyl-tetrahydrofolate (methyl-THF) level, but also the carbon originating from the ADH pathway enters the acetyl-CoA pathway (at formate level). The four electrons gained from methanol oxidation to formate can subsequently be used in the acetyl-CoA pathway to generate acetyl-CoA. Moreover, we do not see any differences in our OD measurements indicating that there is no

increased cell yield when using the MT pathway. Currently, it is also not known if energy conservation in the two pathways is different.

What is the evidence that Desku_2307-9 hydrogenase is confurcating? This argument needs to be better established or the mechanism in Fig. 2 does not necessarily work because the cells would not be able to recycle either the NAD pool or Fd pool to continue growth. Similarly, what is the electron carrier and proposed mechanism for energy conservation when the cells are utilizing the Mta pathway? This is not clear and is potentially related to the point in the previous paragraph. Is the energy conservation expected to be more efficient by not requiring H₂ as an intermediate as suggested by this figure? Addressing this issue will help make the manuscript more accessible to a broader audience. Also, at physiological pH, the relevant species for the terminal reduction product is most likely HS⁻.

There is no experimental/biochemical evidence that the two hydrogenases, Desku_2307-9; 2995-7, are confurcating. However, they were suggested as such by Visser et al. 2013 because of their similarity with the confurcating hydrogenases of Pelotomaculum thermopropionicum. We added this to the text.

Fig. 5A does not add too much, it could be omitted and simply mentioned in the text that the extent of substrate oxidation in the two conditions was similar or moved to supplemental.

We agree that Fig. 5A does not show many differences in the extent of methanol oxidation under the two tested conditions, although again it is visible that the residual methanol concentration in the assay without cobalt is higher. Although this can be said in the text, we believe that it is clearer to the reader to include methanol conversion curves in Fig. 5. Like this, it is easier to associate/relate the profiles of methanol conversion and carbon fractionation.

For Fig. 5B, on what substance is the SCIF being measured? Biomass, methanol? It is not clear from the text at this point and the reader has to wade down to the methods to dig this out. Please specify here.

What was measured was the ¹³C fractionation values of methanol. The legend was adjusted to clarify this.

I don't necessarily agree with the "strongly" in l. 192 because the methanol concentration in the SCIF experiment was very high. Could it be that the ADH pathway is primarily for rapid detoxification and happens to help when cobalt is not present? How realistic is 40 mM methanol in an environment where it is being abiotically produced from CO₂ and H₂? Was methanol detected in the geothermal reservoir where *D. kuznetsovii* was isolated from? If so, what concentration?

We removed the word "strongly". It is possible that ADH pathway is used in response to higher methanol concentrations. What we can say from the results is that ADH pathway does not operate for low methanol concentrations (higher methanol residual concentrations in assays without cobalt). From the fractionation assays, we could only detect $\delta^{13}C$ in assays with cobalt and only when about 60% of the added methanol had been consumed. This seems to indicate that in a first phase only ADH pathway was active, and when methanol concentrations decreased MT started to have a role too in the conversion.

Currently, it is not well known which methanol concentrations can be reached in the deep subsurface and even how methanol is formed (Otomoto et al. 2013). In the few studies we found the concentration in deep subsurface environments was below 1 mM (Otomoto et al. 2013; Yanagawa et al. 2016). Most importantly, these few studies show that a methanol formation and oxidation cycle occurs in the deep subsurface. The effect of temperature and pressure on this cycle is not known. Unfortunately, with the current information we can only speculate.

Yanagawa K, Tani A, Yamamoto N, Hachikubo A, Kano A, Matsumoto R, Suzuki Y (2016). Biogeochemical cycle of methanol in anoxic deep-sea sediments. *Microbes Environ* 31(2):190-3.

Ohtomo Y, Ijiri A, Ikegawa Y, Tsutsumi M, Imachi H, Uramoto G, Hoshino T, Morono Y, Sakai S, Saito Y, Tanikawa W, Hirose T, Inagaki F (2013) Biological CO₂ conversion to acetate in subsurface coal-sand formation using a high-pressure reactor system. *Front Microbiol* 4:361.

The environmental implications suggest that these *D. kuznetsovii* can compete with poorer cobalt scavenger *Moorella thermoacetica*, but what about with a methanogen that the authors suggest should be even better than *M. thermoacetica*? This would suggest that *D. kuznetsovii* would really "win" only in a very low cobalt, but high methanol environment. Do such environments exist in nature? Cobalt can exist as cobalt sulfide, though the melting point is quite low. Does driving a system to high sulfide content deprive every microbe in the system of cobalt and could *D. kuznetsovii* potentially end up creating this niche by its own metabolism?

In the presence of cobalt Moorella thermoacetica grows faster than Desulfotomaculum kuznetsovii. However, in chemostat studies performed with low methanol concentrations (in the presence of sulfate and cobalt) D. kuznetsovii outcompeted M. thermoacetica (Goorissen et al 2004). Such competition studies were never done in media without cobalt, but as the acetogen would require cobalt, there is no competition in the absence of cobalt. In the presence of cobalt, but with limitation of sulfate, D. kuznetsovii can grow in syntropic association with the methanogen.

Goorissen HP1, Stams AJM, Hansen TA (2004) Methanol utilization in defined mixed cultures of thermophilic anaerobes in the presence of sulfate. *FEMS Microbiol Ecol* 49(3):489-94.

One of the questions to consider in the review is "...do you feel that the paper will influence thinking in the field?" This is difficult. *D. kuznetsovii* will set a precedent, but methanol oxidizing sulfate reducing bacteria are not necessarily on every microbiologist's radar. Many will consider them "boutique" and not broadly applicable. So, in the field of C1 metabolism, this will have an effect, but I am uncertain about broader audiences. I think the argument can be made, but was not made effectively in this version.

There is quite some public attention for possible extraterrestrial life. Deep subsurface environments are excellent models to study how chemolithotrophic microbial life on earth and other planets and moons of planets has evolved. Insight into the physiology of bacteria isolated from the deep subsurface is important as they are able to survive and grow at energy and nutrient-limiting conditions. In this respect, it is important to mention that D. kuznetsovii has an interesting physiology. It is a moderate thermophilic anaerobe that can grow chemolithoautotrophically, can fix nitrogen and does not require any vitamins for growth. In addition, it forms the most heat-stable spores ever described for microbes. Spores are not only heat-stable but also resistant to other harsh conditions, such as UV light.

Minor

There are a number of awkward sentences, phrasings, and misspellings. I have pointed out a few here, but I suggest that the editor help the authors to clean this up before any publication.

I. 46 Why is "extremely" in parentheses? Are these significantly more heat resistant than a garden variety endospore from *Bacillus* spp.? *Yes, they are indeed significantly more heat resistant, see O'Sullivan, L. A. et al. (ref 8).*

I. 79 Importnatly -> Importantly *Corrected.*

II. 143-144 "divides two big clusters" is awkward. Perhaps "displays two major clades" *Corrected.*

Fig. 3 Why not use the locus tag in this figure for the *D. kuznetsovii* protein as in other figures? *We agree with the reviewer and added the locus tag to the figure.*

I. 177 allowing to differentiate both pathways -> allowing the pathways to be differentiated *Corrected.*

II. 259-260 Without staining, imaging, and comparison, I'm not sure how this serves as a quality control. What parameters or qualities were used to judge if a sample was of sufficient quality to proceed?

Quality of the samples were judge based on protein concentration (measured with BCA kit). SDS gel was used to compare the profiles of different samples and replicas.

I. 261 120 mV? Did you mean V here? *Changed mV to V.*

II. 263-265 What is the purpose of this step? Why not proceed directly from the liquid sample? *In the new proteomics analysis, samples were run in SDS gel (55 min) to separate proteins; each gel lane was subsequently sliced into 4 sections each of them run in LC-MS/MS. This was done to increase the number of protein identifications.*

I. 273 What was the electrospray voltage? *With the new proteomics analysis and its material and methods this does not apply any more.*

I. 279 ThermoFischer->Thermo Fisher ? *Corrected.*

Answers to Reviewer #4:

The manuscript proposed by Visser, Pieterse, Pinkse, and co-authors describes the proteomic analysis of *Desulfotomaculum kuznetsovii*, isolated from a 3000 m deep geothermal water reservoir and described in 1988. The genome analysis of *D. kuznetsovii* strain 17(T) was published in 2013 by the same research team. Here, the authors focused their attention on the key players involved in methanol metabolism by comparing cells grown with methanol and cells grown with other substrates. Their results show that two methanol degrading pathways are used if considering induced pathways. Furthermore, isotope fractionation allows deciphering the role of each of these pathways at low and high methanol concentrations. I found the manuscript mainly descriptive as it relies only on proteomics and isotope mass spectrometry experiments, and phylogenetic analysis. As per the editorial request, I am commenting on the proteomic analyses included in this report.

The proteomic conclusions are based on only two biological replicates per condition and semi-quantitation of proteins by the spectral count approach. The raw data are not available and it is difficult to judge the statistics of these datasets. Central metabolism is modulated quite drastically in bacteria and the authors could easily point at the key proteins involved in methanol metabolism. In this sense, I am expecting a more in-depth characterization of the corresponding enzymes than just a phylogenetic tree.

1. The comparative proteomic data analysis comprises four different conditions but only two biological replicates. This gives a rather low confidence in these results. Furthermore, the authors mentioned (line 96) that the lactate growth condition was used as a reference. Table S2 shows that this reference is not of quality, with 6469 MS/MS spectra assigned for the first replicate and 8624 MS/MS spectra assigned for the second replicate. The second replicate dataset is thus inflated of +33%! Usually, a normalization of the protein material should be done for comparing equal quantities of material. Here, the reference shows unusual variations. I recommend to the authors to consider new biological replicates and precise normalization of their samples before MS/MS recording.

Proteomic measurements have been repeated using biological triplicates/quadruplates and an MS1 intensity based quantitation of peptides and proteins using MaxQuant algorithms.

2. Another problem appears when the different conditions are compared. In the MeOH -B12 condition (replicates 1 and 2), more MS/MS spectra are recorded (in average +17%) compared to the MeOH condition (replicates 1 and 2). The authors should normalize the protein quantities before peptide extraction and recording their tandem mass spectrometry datasets.

MaxQuants LFQ normalization algorithm has now been used and conditions are compared using volcano plots.

3. Protein abundances have been quantified based on MS/MS spectral count and not peptide count. The authors should modify their manuscript accordingly. In addition, such measure is valid if the tandem mass spectrometer has enough time to record several spectra per peptides if abundant. The authors performed a 60 min gradient with a dynamic exclusion time set to 30 seconds. This gradient is rather short for a comprehensive view of a proteomic sample, but I expect that a dynamic exclusion of no more than 10 sec is activated for spectral count. Therefore, only the most abundant proteins will be quantified with enough spectral counts for comparative proteomics. I found these settings not relevant here.

In the revised version we quantify based on MS1 intensity which is not influenced by the dynamic exclusion time setting.

4. Table S2 shows only a partial view on the proteins identified. Some ambiguities are noted: i) Proteins that are 3 times more abundant in one condition as indicated in the manuscript, or proteins with at least 4 peptide counts as mentioned in Table S2? Table S2 lists 626 proteins while 892 have been identified with at least two different peptides. Selecting and presenting only partial proteomic data is really unusual. I wonder what's the reason...

Vulcano plot presented in the revised version show all proteins quantified. In supplementary file S2 all the proteins that were found to be significantly different between the different conditions can be seen.

5. Another point is the growth of the bacteria. The authors mentioned that cells were harvested at the late exponential phase. How this growth was followed? Are the authors sure that the same growth stage have been compared? Is this based on a simple optical density measurement, or a sound kinetic analysis with growth curve and equivalent growth stages?

We agree that the material and methods was not clear enough regarding how growth was monitored. We used optical density measurements and methanol, sulfate and sulfide concentration measurements. We adjusted the text of the materials and methods section accordingly.

6. As recommended by the proteomics community, the raw data should be deposited in a public repository where reviewers may have access for checking their quality.

Proteomics data are available via PRIDE Project Accession PXD006899.

7. Figure 1 shows a heatmap of selected proteins. The authors did not use the lactate reference, but rather proposed a peptide count of all conditions for calculating the fold changes. This is rather unusual. The statistics to validate these fold changes are not presented and are not explained. A p value for each protein fold change should be available, as well as a Benjamin-Hochberg post-hoc test.

Conditions each measured in triplo or quadruplo have now been compared using T-tests of which the results are being visualized in a volcano plot in figure 1 (and all values in supplementary file S2).

8. Some predicted functions are surprisingly different between Table S2 and Figure 1 (see for example, the Desku_1072 and Desku_0050 protein labels).

This table and figure have been replaced with the new proteome data. The predicted functions are now similar in S2 and figure 1.

9. The analysis of the protein dataset is superficial. For example, Desku_0057 is annotated as a ferredoxin and is highly abundant (as much as MtaA). In fact, a simple BLAST analysis shows that this

protein is not a ferredoxin but rather a multi-domain protein including an Fe-S ferredoxin-like domain at its N-terminal.

For the predicted function/name of the proteins we used the annotation of the genome. We agree with the reviewer that annotations can be superficial. However, the aim of this study was not to correct superficial annotated genes/proteins.

10. The study could be complemented with comparative genomics of the whole pathways, rather than phylogenetic trees.

We wanted to specifically focus on the evolution of MTA and ADH among different phylogenetic groups. We believe that these genes are the best targets for that. A genomic comparison of the whole pathways could bring other elements but would not highlight differences in the targeted genes. In addition, the databases for genes and proteins are much larger, they contain more species than a database of whole genomes, which is needed for comparative pathway analysis, and therefore might miss essential information.

11. The authors cited a reference from 2002 (ref 51) to support their LC-MS/MS analysis. This reference has nothing related to the instrument used here, a Q-Exactive Plus tandem mass spectrometer.

References concerning proteomics measurements and analysis have been changed accordingly.

12. The general quality of the figures and the supplementary material is low. How these data are normalized? Where are the fold changes and associated p values for comparing these results?

The different growth conditions measured in triplo or quadruplo are now being compared using T-tests of which the results are being visualized in a volcano plot in figure 1 (and all values in supplementary file S2).

13. I wonder if it is the strain 17(T) that has been analyzed by proteomics? This is not mentioned in the manuscript.

The strain used for the entire study is the type strain, strain 17(T). We included this information in the abstract and main text.

Reviewer #2 (Remarks to the Author):

The manuscript of Drs. Sousa and colleagues describes data supporting the presence of two enzyme systems for methanol degradation by *Desulfotomaculum kuznetsovii*. A strong argument is made for the competitive advantage provided by these alternative pathways for substrate oxidation available to this bacterium. One pathway is dependent on a corrinoid-dependent methyl transferase step while the second appears to involve an alcohol dehydrogenase activity that is independent of the corrinoid. Significantly different stable isotope fractionations were obtained in the presence of cobalt (needed for the corrinoid –dependent pathway) versus in the absence of cobalt which should prevent the synthesis of the corrinoid. The growth of *D. kuznetsovii* on methanol occurred under both medium conditions. Proteomics studies supported the suggestion that candidate enzymes needed were increased in expression with methanol as substrate and in the presence of cobalt.

The paper is well written and the data presented clearly. The authors should respond to a few points of confusion.

A) The cultures for proteomics were grown to late exponential phase. Although ribosomal proteins were not significantly decreased in abundance, would it not be expected that any substrate limiting continued exponential growth might be reflected in the protein abundances obtained?

Alternatively, the entry into late exponential phase might also be a response to inhibitors such as sulfide concentrations. Are the two methanol degradation systems equally resistant to sulfide?

B) Catalase also increased in abundance at low methanol concentrations. Were there other evidences of oxidative stress?

C) The lag times for isotope fractionation when methanol was at higher concentrations and the inability of the alcohol dehydrogenase to oxidize methanol at low concentrations were taken as evidence that both enzyme systems were functional when there was methanol and cobalt present. However, what were the limits of methanol concentration changes detectable when 20 mM was present? What was the limit in the detection of isotope fractionation? Were these measurements sufficiently sensitive to see the small changes expected?

D) Did the authors ever measure the actual cobalt concentration in the medium when cobalt was limiting?

E) In the introduction to hydrogenases of this bacterium, could you also provide their metal content? Do any of these have signal sequences for transfer to a periplasmic location or are all cytoplasmic?

F) The paragraph describing the genomic content of other methanol-degrading SRBs seems a bit distracting. Consider omitting lines 240-245.

G) Line 201: suggest... 'the confurcating hydrogenases'.

H) Line 158: Is 'a remarkable and unexpected finding' meant? Or is 'a remarkably unexpected finding' meant?

I) Line 256: dehydrogenases

J) Line 342: Please indicate what is meant by "normal" logarithm.

K) Figure 1: Please indicate the log base 2 (if accurate) on the x-axis of panel A. In panel B, please indicate what the colors mean. Clearly they are used differently in the two panels. It would be clearer if the colors in the two panels were actually completely different. In panel B, log based 10 numbers resulting from the LFQ analysis are not often seen for protein abundances. More information about the calculation yielding this value would be helpful. That would allow the number values to be better understood. Clearly the color key has reflected the average of the numbers shown.

L) In *Salmonella*, the number of genes for B12 biosynthesis is somewhere near 30. Are all the enzymes for corrinoid biosynthesis increased in expression when cobalt is present with methanol? There were several precorrinoid enzymes that seemed not to be increased.

Reviewer #3 (Remarks to the Author):

In this resubmission, I feel that the authors have addressed most of the initial concerns. I am not completely confident that the issues with peptide quantification have been addressed, but coupling the isotope fractionation data with the observations here are fairly compelling.

Minor outstanding grammatical/style suggestions, a non-exhaustive list:

- l. 39 ...high temperatures and oligotrophic conditions often prevail.
- l. 40 "Resident microbial communities" instead of "The ambient microbes"?
- l. 46 I think the parentheses are unnecessarily distracting. Either they are or aren't extremely heat resistant relative to others.
- l. 64 The methanol metabolism of sulfate reducing bacteria (SRB) has not been extensively studied.
- l. 74 insert "putative" or "predicted" after "revealed"
- l. 105 "a methanol" -> "a predicted methanol"
- l. 111-122 This paragraph could be condensed a bit by eliminating redundant statements.
- l. 142 Delete "different types of anaerobic microorganisms, such as"? If the reader has made it this far, they probably appreciate these are anaerobes.
- l. 147 I would delete the clause after the comma. Horizontal gene transfer in the evolution of microbial genomes is neither uncommon or surprising.
- l. 176 significantly
- l. 202 can -> may, until the experiments are done, some qualification is justified.
- l. 214 delete "is the cobalamin binding subunit of the methyltransferase, which"

Reviewer #4 (Remarks to the Author):

The manuscript NCOMMS-16-25212C describes the proteomic analysis of *Desulfotomaculum kuznetsovii* grown in various media. The results are crystal clear, showing that two methanol degrading pathways are used, depending on methanol concentration. The authors have performed additional proteomics analysis to reinforce their conclusions and deposited their raw files in the PRIDE repository in a partial submission mode (only the 84 raw data files and the MaxQuant results are available). Based on state-of-the-art XIC data analysis with the MaxQuant software the proteomic conclusions are robust.

Minor comments:

1) P14. The sentence "... with a contaminants database that contains sequences of common contaminants like Trypsins (P00760, bovin and P00761, porcin) and human keratins (Keratin K22E (P35908), Keratin K1C9 (P35527), Keratin K2C1 (P04264) and Keratin K1CI (P35527))." could be shortened into "...with a most common contaminants database."

2) P15. The authors mentioned that "Accepted were peptides and proteins with a false discovery rate (FDR) of less than 1% and proteins with at least 2 identified peptides of which at least one should be unique and at least one should be unmodified." The sentence is unclear whether the FDR applies to peptide or proteins, or both... Furthermore, the maxquant results available in the PXD006899 dataset (file proteinGroups.txt) indicate a series of proteins identified with only one peptide (for example, proteins F6CGU8 and F6CGY7). The authors could clarify this point by mentioning how much protein groups were really validated with at least 2 identified peptides.

3) Strangely, the authors did not mention in their revised version the PRIDE accession number for

their proteomic dataset.

4) In the acknowledgement section, the authors thank Dr. M. Kellmann and Dr. T. Moehring for their assistance in the Q Exactive Plus analyses. However, no data recorded with such instrument are commented in the manuscript. Is there any error?

We would like to thank again all reviewers for their feedback. Below you can find detailed responses to all the questions (text in black italics).

Answers to Reviewer #2:

The manuscript of Drs. Sousa and colleagues describes data supporting the presence of two enzyme systems for methanol degradation by *Desulfotomaculum kuznetsovii*. A strong argument is made for the competitive advantage provided by these alternative pathways for substrate oxidation available to this bacterium. One pathway is dependent on a corrinoid-dependent methyl transferase step while the second appears to involve an alcohol dehydrogenase activity that is independent of the corrinoid. Significantly different stable isotope fractionations were obtained in the presence of cobalt (needed for the corrinoid – dependent pathway) versus in the absence of cobalt which should prevent the synthesis of the corrinoid. The growth of *D. kuznetsovii* on methanol occurred under both medium conditions. Proteomics studies supported the suggestion that candidate enzymes needed were increased in expression with methanol as substrate and in the presence of cobalt.

The paper is well written and the data presented clearly. The authors should respond to a few points of confusion.

A) The cultures for proteomics were grown to late exponential phase. Although ribosomal proteins were not significantly decreased in abundance, would it not be expected that any substrate limiting continued exponential growth might be reflected in the protein abundances obtained? Alternatively, the entry into late exponential phase might also be a response to inhibitors such as sulfide concentrations. Are the two methanol degradation systems equally resistant to sulfide?

With the term 'late exponential' we wanted to refer to a later phase of the exponential phase but not yet when growth rate starts to decrease. We agree this term is not the most correct and may be misleading. Cultures were harvested when approximately 70-80% of the substrate was consumed. This is now indicated in the manuscript in L287-288.

Regarding the inhibitory response caused by sulfide concentrations, this does not occur until after degrading approximately 25 mM of methanol, as described in L333-335. Moreover, in L265 we describe that for the proteomics growth conditions we did not use more than 20 mM of methanol.

B) Catalase also increased in abundance at low methanol concentrations. Were there other evidences of oxidative stress?

This is a good point and we do not have a full explanation for this. However, as lower amounts of sulfide are produced with low concentrations of methanol the redox potential of the medium will differ which may cause the expression of the catalase gene. This became apparent while processing the samples as the ones with low sulfide became pink because of the change in the redox indicator (rezasurin).

C) The lag times for isotope fractionation when methanol was at higher concentrations and the inability of the alcohol dehydrogenase to oxidize methanol at low concentrations were taken as evidence that both enzyme systems were functional when there was methanol and cobalt present. However, what were the limits of methanol concentration changes detectable when 20 mM was present? What was the limit in the detection of isotope fractionation? Were these measurements sufficiently sensitive to see the small changes expected?

The 'limits in the detection of isotope fractionation' depend on the uncertainty of the used isotope analysis technique and the number and quality of samples used for fractionation analysis. Practically, the correlation coefficient and alignment of the regression line and the magnitude of the 95% confidence interval tell us whether the data are significant or not. As stated in the manuscript, the uncertainty of carbon stable isotope analysis of methanol was usually less than 0.4 ‰.

D) Did the authors ever measure the actual cobalt concentration in the medium when cobalt was limiting?

Cultures were transferred at least five times, with 1% inoculum, to guarantee complete depletion of Co from the medium and cells. Considering dilution concentration of cobalt goes from 0.5 μM in the normal medium to 0.5×10^{-10} μM in the medium without cobalt. Since we performed an additional proteomics analysis for the previously submitted version we used cultures that were transferred at least 10 times (0.5×10^{-20} μM). Because 10^{-10} μM Co is far below the detection limit of the available equipment we did not consider measuring this.

E) In the introduction to hydrogenases of this bacterium, could you also provide their metal content? Do any of these have signal sequences for transfer to a periplasmic location or are all cytoplasmic?
In D. kuznetsovii all hydrogenases are cytoplasmic. These were described by Visser et al. 2015 (reference 7 in the manuscript).

F) The paragraph describing the genomic content of other methanol-degrading SRBs seems a bit distracting. Consider omitting lines 240-245.
This paragraph was deleted.

G) Line 201: suggest... 'the confurcating hydrogenases'.
Corrected.

H) Line 158: Is 'a remarkable and unexpected finding' meant? Or is 'a remarkably unexpected finding' meant?
Corrected to 'a remarkably unexpected finding'.

I) Line 256: dehydrogenases
Corrected.

J) Line 342: Please indicate what is meant by "normal" logarithm.
Thank you for this correction we actually did all the calculations using a 10-base logarithm. This was now clearly indicated in the text and figures.

K) Figure 1: Please indicate the log base 2 (if accurate) on the x-axis of panel A. In panel B, please indicate what the colors mean. Clearly they are used differently in the two panels. It would be clearer if the colors in the two panels were actually completely different. In panel B, log based 10 numbers resulting from the LFQ analysis are not often seen for protein abundances. More information about the calculation yielding this value would be helpful. That would allow the number values to be better understood. Clearly the color key has reflected the average of the numbers shown.
Colours of Figure 1B were changed and a colour legend included. We used a \log_{10} for all the calculations in the manuscript. The colour shown in panel B is for each of the individual replicas and not averaged numbers.

L) In Salmonella, the number of genes for B12 biosynthesis is somewhere near 30. Are all the enzymes for corrinoid biosynthesis increased in expression when cobalt is present with methanol? There were several precorrinoid enzymes that seemed not to be increased.
The reviewer is right to say that there are more B12 biosynthesis genes present in the genome of D. kuznetsovii (for example Desku_1459-1466) and that they are not increased in abundance in the presence of cobalt and methanol. The increased abundance is only the case for genes part of the operon structure Desku_0050-0060. We changed the text accordingly, L115-118.

Answers to Reviewer #3:

In this resubmission, I feel that the authors have addressed most of the initial concerns. I am not completely confident that the issues with peptide quantification have been addressed, but coupling the isotope fractionation data with the observations here are fairly compelling.

Proteomic measurements have been repeated completely using biological triplicates/quadruples and an MS1 intensity based quantitation of peptides and proteins using MaxQuant algorithms.

Protein quantitation is therefore based on MS1 intensity (taken as the average of the 3 highest measured points) which is not influenced by the dynamic exclusion time setting.

A direct comparison between 2 conditions is made. The proteins normalized LFQ intensities obtained for the different conditions are being compared using T-tests and plotted as volcano plots with (natural) Log Protein abundance ratio (conditionA/ConditionB) versus Log p-values as in Figure 1A.

An overview of 4 conditions is given in Figure 1B which, for a selected number of proteins, shows the natural logarithm of the protein abundances observed as a number together with a colour indication.

Minor outstanding grammatical/style suggestions, a non-exhaustive list:

I. 39 ...high temperatures and oligotrophic conditions often prevail. *Corrected.*

I. 40 "Resident microbial communities" instead of "The ambient microbes"? *Corrected.*

- I. 46 I think the parentheses are unnecessarily distracting. Either they are or aren't extremely heat resistant relative to others. *Parentheses were removed.*
- I. 64 The methanol metabolism of sulfate reducing bacteria (SRB) has not been extensively studied. *Corrected.*
- I. 74 insert "putative" or "predicted" after "revealed" *Inserted.*
- I. 105 "a methanol" -> "a predicted methanol" *Inserted.*
- I. 111-122 This paragraph could be condensed a bit by eliminating redundant statements. *Redundancies were deleted.*
- I. 142 Delete "different types of anaerobic microorganisms, such as"? If the reader has made it this far, they probably appreciate these are anaerobes. *Deleted.*
- I. 147 I would delete the clause after the comma. Horizontal gene transfer in the evolution of microbial genomes is neither uncommon or surprising. *The unexpected fact is that the MT system of D. kuznetsovii is evolutionarily closer to the MT system of methanogens than to that of acetogens.*
- I. 176 significantly *Corrected.*
- I. 202 can -> may, until the experiments are done, some qualification is justified. *Replaced.*
- I. 214 delete "is the cobalamin binding subunit of the methyltransferase, which" *Deleted.*

Answers to Reviewer #4:

The manuscript NCOMMS-16-25212C describes the proteomic analysis of *Desulfotomaculum kuznetsovii* grown in various media. The results are crystal clear, showing that two methanol degrading pathways are used, depending on methanol concentration. The authors have performed additional proteomics analysis to reinforce their conclusions and deposited their raw files in the PRIDE repository in a partial submission mode (only the 84 raw data files and the MaxQuant results are available). Based on state-of-the-art XIC data analysis with the MaxQuant software the proteomic conclusions are robust.

Minor comments:

1) P14. The sentence "... with a contaminants database that contains sequences of common contaminants like Trypsins (P00760, bovin and P00761, porcin) and human keratins (Keratin K22E (P35908), Keratin K1C9 (P35527), Keratin K2C1 (P04264) and Keratin K1CI (P35527))." could be shortened into "...with a most common contaminants database."

The text was shortened according to the suggestion of the reviewer.

2) P15. The authors mentioned that "Accepted were peptides and proteins with a false discovery rate (FDR) of less than 1% and proteins with at least 2 identified peptides of which at least one should be unique and at least one should be unmodified." The sentence is unclear whether the FDR applies to peptide or proteins, or both... Furthermore, the maxquant results available in the PXD006899 dataset (file proteinGroups.txt) indicate a series of proteins identified with only one peptide (for example, proteins F6CGU8 and F6CGY7). The authors could clarify this point by mentioning how much protein groups were really validated with at least 2 identified peptides.

From the original 1622 protein groups in the original MaxQuant result, 208 were filtered out leaving 1414 protein groups (added in L322-323).

3) Strangely, the authors did not mention in their revised version the PRIDE accession number for their proteomic dataset.

The following information has been inserted in the 'Methods' section:

"The mass spectrometry proteomics data have been deposited to the ProteomeXchange Consortium via the PRIDE⁵⁸ partner repository with the dataset identifier PXD006899".

*58. Vizcaino, J.A., Csordas, A., del-Toro, N., Dianes, J.A., Griss, J., Lavidas, I., Mayer, G., Perez-Riverol, Y., Reisinger, F., Ternent, T., Xu, Q.W., Wang, R. & Hermjakob, H. Update of the PRIDE database and related tools. *Nucleic Acids Res.* 44, D447-D456 (2016). (PubMed ID: 26527722).*

4) In the acknowledgement section, the authors thank Dr. M. Kellmann and Dr. T. Moehring for their assistance in the Q Exactive Plus analyses. However, no data recorded with such instrument are commented in the manuscript. Is there any error?

The reviewer is correct. They have been involved in a previous proteomics analysis that had to be repeated. They did not contribute for the data included in this version of the manuscript.